# AutoFloorplan: Evolving Heuristics for Chip Floorplanning with Large Language Models and Textual Gradient-Guided Repair

## Abstract

Chip floorplanning is the cornerstone of modern Very Large Scale Integration (VLSI) design, but it remains an impenetrable NP-hard combinatorial optimization problem. Designing effective heuristic algorithms to explore its large solution space under complex constraints is a challenging task that traditionally relies on rich human expertise. In this study, we propose AutoFloorplan, a novel evolutionary learning framework that automatically discovers complex heuristics floorplanning algorithm. We utilize Large Language Models (LLMs) as intelligent population generators capable of creating diverse heuristics floorplanning algorithm expressed as code. However, a fundamental challenge is that many LLM-generated heuristics are invalid and do not conform to the strict geometric and topological constraints of floorplanning. To address this problem, we design a novel repair operator based on textual gradients. This operator analyzes the causes of inefficiencies in the generated heuristics and provides corrective feedback to steer the algorithmic structure towards effective and high-performance alternatives. Our framework significantly improves the speed of discovering legitimate and effective heuristics and iterating on algorithm performance. Extensive experiments on eight different public circuits show that AutoFloorplan outperforms current State-of-the-Art floorplanning algorithms. The code of AutoFloorplan can be found at `https://anonymous.4open.science/r/AutoFloorplan-main`.

## 1 Introduction

Chip floorplanning is a fundamental and critical phase in the physical design of modern VLSI circuits, where functional blocks are arranged on a chip to minimize area and wire length while meeting complex geometrical and topological constraints Kahng et al. (2011); Weste & Harris (2015); Chen & Chang (2006). As an NP-hard combinatorial optimization problem with an enormous and intricate solution space Hartmanis (1982), it sets a prototype for downstream tasks and determines an upper bound on the final PPA (power, performance, area). Over the decades, methods such as sequence pair representations Murata et al. (1996); Liu et al. (2008); Wang et al. (2007), explored via metaheuristic search algorithms like simulated annealing and evolutionary algorithms, have been widely used; however, their effectiveness depends heavily on hand-crafted heuristic components that require substantial domain expertise and complex engineering.

To reduce the burden of manual heuristics algorithm, researchers have increasingly focused on automatic heuristics design (AHD). EAs have emerged as powerful paradigms for automated problem solving, capable of evolving solutions to complex problems without explicit programming Eiben & Smith (2015). More recently, the emergence of Large Language Models (LLMs) has revolutionized the field of artificial intelligence, demonstrating an unprecedented ability to understand and generate human-like text, especially with respect to source code Mo et al. (2025); Chen et al. (2021); Liu et al. (2024). The potential to leverage LLMs to generate optimization heuristics directly in the form of executable code provides an exciting new frontier that promises to create more complex and semantically rich policies than traditional methods Romera-Paredes et al. (2024); Liu et al. (2024).

Despite the great potential of large language models, directly applying their generated heuristic rules to chip floorplanning still faces fundamental challenges: the domain is governed by strict and non-

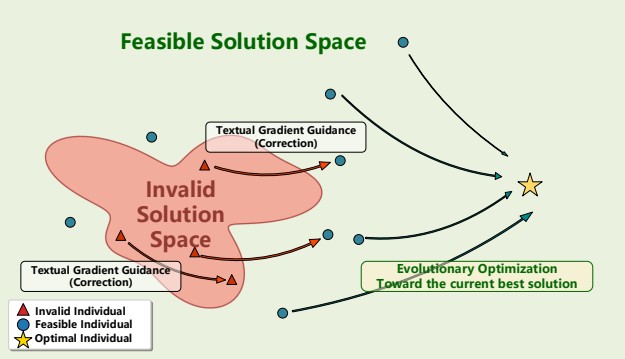

Figure 1: Schematic illustration of the principle of synergy between evolutionary search and textual gradient.

negotiable constraints, causing many LLMs-generated heuristic algorithms to violate these rules and fail to produce valid floorplans that satisfy the required geometric and topological conditions. In existing generate-and-filter evolutionary Romera-Paredes et al. (2024); Liu et al. (2024); Yao et al. (2025) frameworks, such invalid algorithmic individuals are typically discarded outright, which not only wastes multiple iterations (computational resources) but also leads to an insufficient number of valid individuals, resulting in slow convergence or even failure of the evolutionary process.

In order to bridge this gap, this paper proposes a novel evolutionary learning framework, AutoFloorplan. Specifically, as shown in the schematic diagram in Figure 1, AutoFloorplan does not simply follow the discard-restart evolutionary model, but propose a more efficient and intelligent corrective learning mechanism. We think that every time "drop and start again", will be able to learn from our mistakes since that every invalid generation contains a valuable learning signal. When an LLMs-generated floorplanning algorithm is invalid due to a constraint violation, AutoFloorplan does not eliminate it outright. Instead, the system deeply analyzes the root cause of the failure and translates it into structured and instructive corrective feedback. This feedback, like the gradient in text space, is sent back to the LLM, guiding the model precisely how to modify the structure and logic of the algorithm in order to bypass the current trap. The more reasonable way we treat to the invalid outputs comparing with simple discarding them greatly improves the efficiency and success rate of the algorithmic discovery, allowing the model to converge faster to the region of the solution space that is both legitimate and high-performance. Our contribution is reflected in three aspects:

- A new algorithm design paradigm is introduced that integrates evolutionary learning with a textual gradient-guided repair mechanism, which identifies why heuristics fail and provides targeted natural-language corrections—significantly improving constraint handling in LLMs-driven floorplanning.

- Unique use of LLMs as an intelligent floorplanning algorithm generator. This allows us to create diverse algorithm of floorplanning heuristics and present them directly in code, thus breaking through the limitations of hand-written or simple evolutionary heuristics.

- Extensive experiments are conducted on eight widely used public MCNC and GSRC benchmark circuits. The experimental results show that AutoFloorplan's evolved heuristics consistently provide superior floorplanning solutions, outperforming current state-of-the-art (SOTA) floorplanning algorithms in terms of both area and wire length minimization.

## 2 RELATED WORK

### 2.1 CHIP FROORPLANNING

Recent studies have explored machine learning–based approaches to improve floorplanning effectiveness, including frameworks that integrate graph convolutional networks and reinforcement learning Xu et al. (2021); Liu et al. (2022); Basso et al. (2024), as well as RL-based methods for block floorplanning and local perturbation optimization Mirhoseini et al. (2021); He et al. (2020); Vashisht

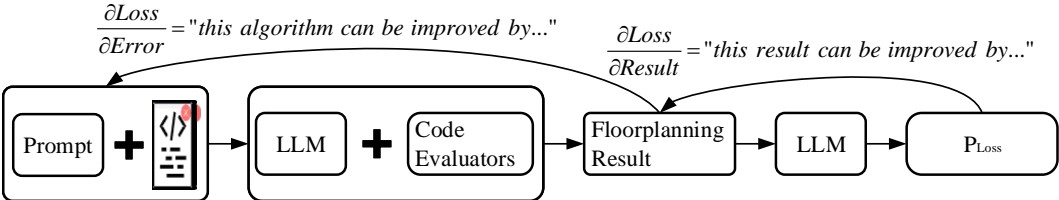

Figure 2: Textual gradient workflow diagram.

et al. (2020); Lai et al. (2023). Despite this progress, classical meta-heuristics such as SA Mostafa et al. (2024), GA Chang & Ting (2022), and ant colony optimization Tang & Yao (2007) remain widely used due to their controllability under strong constraints, stability in complex nonconvex search spaces, lack of dependence on training data, and natural support for multi-objective optimization. However, their heuristic design still requires substantial human expertise, motivating the development of AutoFloorplan to automatically discover effective heuristics floorplanning algorithm.

## 2.2 AUTOMATED ALGORITHM DESIGN

Automated algorithm design has gained significant attention, with studies showing that metaheuristic generation can reduce manual effort Zhao et al. (2023). Approaches such as evolutionary algorithms, genetic programming, and program synthesis enable automatic derivation of optimization strategies Stützle & López-Ibáñez (2018); Bömer et al. (2025). Recent advances in LLMs-driven algorithm discovery—such as FunSearch Romera-Paredes et al. (2024), EoH Liu et al. (2024), PoH Mu et al. (2025), CEoH Bömer et al. (2025), and Yao et al. (2025) — demonstrate that LLMs can generate and iteratively refine heuristics that rival or surpass human-designed methods. However, when applied directly to tightly constrained domains like chip floorplanning, LLMs often generate invalid solutions, wasting computation and missing opportunities for guided improvement. AutoFloorplan addresses this gap by introducing a textual gradient mechanism that analyzes failure cases and transforms them into structured, corrective feedback, enabling more effective learning and constraint handling.

## 3 PRELIMINARIES

Compound AI systems orchestrate multiple black-box components (LLMs, tools, simulators) connected through natural-language interfaces, making end-to-end differentiation impossible with standard backpropagation. As show in Figure 2, TextGrad Yuksekgonul et al. (2025) treats any such system as a computation graph in which nodes are optimizable text variables (prompts, code, molecules, plans, agent trajectories, etc.) and edges are arbitrary forward functions (LLMs calls, APIs, evaluators). A loss node $P_{loss}$ produces scalar or textual feedback.

A textual gradient with respect to variable $x$ is a natural-language critique:

$$\frac{\partial L}{\partial x} = \nabla_{text}(x, y, \frac{\partial L}{\partial y}) = LLM('Given\ the\ forward\ pass\ x \rightarrow y\ and\ criticisms\ on\ y: \tag{1}$$
$$... \ Suggest\ concrete\ revisions\ to\ x\ that\ reduce\ the\ loss.')$$

Textual gradients are computed recursively from the loss backward through the graph (chain-rule analogue) and applied via a textual descent step:

$$x^{(t+1)} = LLM('Revise\ x\ to\ incorporate\ these\ criticisms: ...' + \frac{\partial L}{\partial x}) \tag{2}$$

We directly adopt TextGrad's textual backpropagation and descent procedure, constructing task-specific computation graphs for floorplanning as described in AutoFloorplan. In AutoFloorplan, monotonic improvement is encouraged through the elite-preservation mechanism within the population evolution process: only strategies that outperform previous ones are retained, ensuring incremental progress even though LLMs-based feedback itself is not guaranteed to be strictly monotonic.

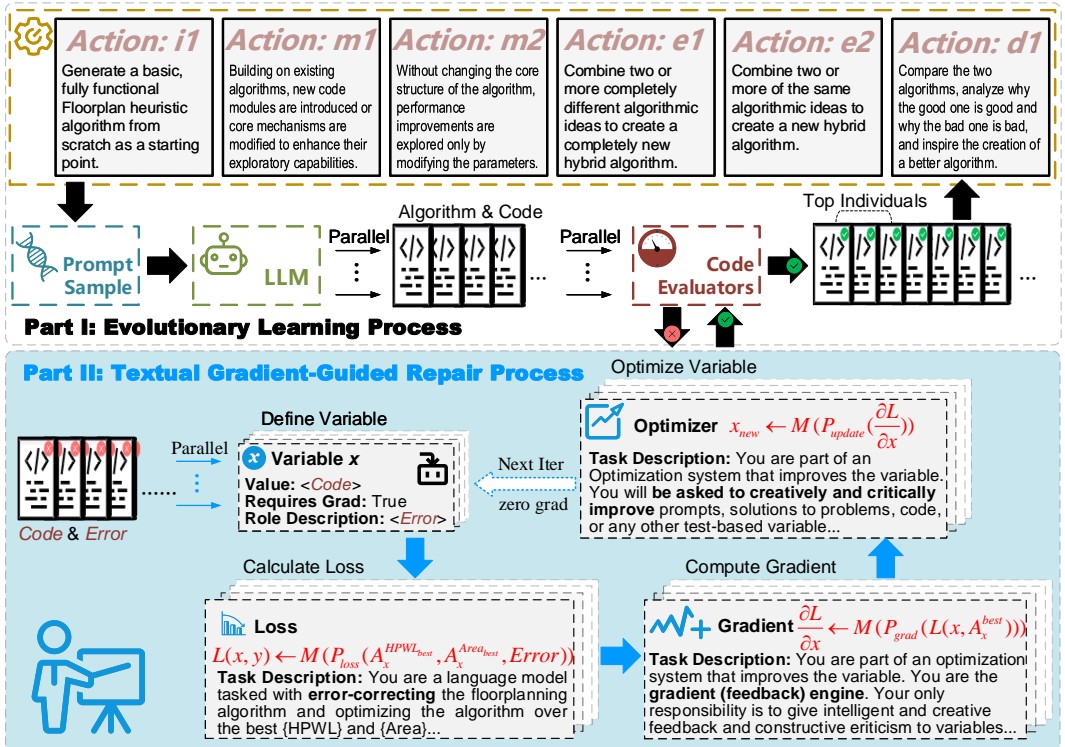

Figure 3: The overview of AutoFloorplan. It can be seen that AutoFloorplan consists of two main parts, one is the evolutionary learning process, which consists of three originals, namely, Prompt operator sampling, LLM, and code evaluators; the other part is the textual gradient-guided repair process.

## 4    AUTOFLOORPLAN

To address the challenges that population-based, LLM-driven algorithm-heuristic design (AHD) methods face when exploring the complex heuristic algorithm space and satisfying valid floorplan constraints in chip floorplanning, we propose AutoFloorplan. As illustrated in Figure 3, AutoFloorplan adopts a two-loop intelligent framework. The outer loop performs evolutionary learning, enabling broad, creative exploration of the algorithmic design space through LLM generation and diverse evolutionary prompt operators. The inner loop provides a textual-gradient–guided repair mechanism, which conducts fine-grained logical correction and optimization by intelligently analyzing and repairing failed algorithmic individuals.

### 4.1    PART I: EVOLUTIONARY LEARNING PROCESSES

The goal of this process is to mimic biological evolution by creating superior performing heuristic floorplanning algorithms from scratch through iterative generation, evaluation, and selection. The following are the technical details for Part I:

#### 4.1.1    LLM-BASED EVOLUTIONARY SEARCH ACTIONS FOR FLOORPLANNING

Evolution strategy (ES) is a classical black-box optimization technique Pošík et al. (2012) that can provide an optimization framework for LLMs in Black-Box environments, thus empowering LLMs with flexible global search capabilities. As shown in Part I of Figure 3, AutoFloorplan combines the advantages of ES and LLMs to design six types of evolutionary prompts (i1, m1, m2, e1, e2, d1) used to design floorplanning heuristics. These operators are briefly described below, Specific prompts are detailed in **Appendix A.2**.

- *Action i1*. Generate a basic, fully functional floorplanning heuristic algorithm from scratch.

- *Action m1.* Introducing new code modules or modifying the core mechanism to enhance its exploration capabilities based on the existing floorplan heuristic algorithm.

- *Action m2.* Without changing the core structure of the existing floorplan heuristic algorithm, performance improvements can only be explored by modifying the parameters.

- *Action e1 / e2.* Analyze the differences / common ideas between two or more floorplan heuristic algorithms and create a new and better floorplan heuristic algorithm based on the execution results of the algorithms.

- *Action d1.* Inspired by algorithms such as DPO Rafailov et al. (2023), two elite individuals are selected for comparative learning in order to directly set LLM preferences. First, two individuals are randomly selected within the elite individuals, respectively. Then, the LLM is asked to compare the floorplan results obtained from the execution of the algorithm and evaluate why the high-scoring individual is superior. Finally, a superior floorplanning algorithm is asked to be designed, accompanied by a short description of the algorithmic idea.

### 4.1.2 LLMS

In AutoFloorplan, a wide variety of LLMs, such as ChatGPT Ouyang et al. (2022), can be flexibly integrated. These LLMs can be open source models, proprietary commercial products or fine-tuned variants for specific tasks or domains. This pervasiveness allows AutoFloorplan to leverage the power of LLMs in generating, reasoning about and optimizing algorithm.

### 4.1.3 CODE EVALUATORS

At AutoFloorplan, code evaluators are used to evaluate the resulting floorplanning algorithms, and the evaluators score (i.e., calculate the "fitness" of) each algorithm based on key performance indicators such as wire length and chip area:

$$Score(x) = \frac{A_x^{HPWL_{current}}}{A_x^{HPWL_{best}}} + \frac{A_x^{Area_{current}}}{A_x^{Area_{best}}} \tag{3}$$

where $A_x^{HPWL}$, $A_x^{HPWL_{best}}$, $A_x^{Area}$, $A_x^{Area_{best}}$ are the results of the current algorithm and the optimal algorithm execution so far, respectively.

In addition to this, the legitimacy of the floorplan is evaluated, such as 'evaluate if blocks are out of the chip outline', 'if there is any overlap in blocks' and 'if all the blocks are completely placed'(Case study in **Appendix A.7**). Furthermore, Appendix A.1 summarizes the errors that occur in the heuristic floorplanning algorithm generated by the LLMs. Algorithms that violate the constraints and the corresponding Error will be sent to the textual gradient-guided repair loop to complete the repair.

The entire execution of AutoFloorplan is highly parallelized, from the parallel response of LLM to the parallel evaluation of the floorplanning heuristic algorithm in different threads. In addition, for efficiency, we set a timeout duration (timeout duration = 1800s). If the generated heuristic algorithm fails to complete floorplanning within this timeout duration, the process is terminated and the algorithm is discarded without entering the elite population and repair procedure.

### 4.1.4 EVOLUTIONARY PROCESS

To ensure stable and continuous improvement during the evolutionary search, AutoFloorplan adopts an elite population strategy. After each generation of heuristic floorplanning algorithms is produced and evaluated, all valid individuals are ranked according to their fitness score calculated using formula 3. The top-performing individuals—referred to as the elite population—are directly preserved into the next generation without modification.

This strategy serves two purposes. First, it guarantees that the best solutions found so far are never lost due to stochastic variations introduced by LLMs-based generation, prompt operators, or textual-gradient repairs. Second, elite individuals act as high-quality anchors that guide the evolutionary process toward more effective regions of the heuristic algorithm design space.

Although theoretical properties of textual gradients (e.g., directionality or convergence guarantees) remain unclear, we find that combining textual-gradient–guided repairs with elite preservation yields

consistent empirical gains. By retaining strong individuals and repairing lower-performing ones, AutoFloorplan maintains diversity while ensuring that the population steadily progresses toward better algorithmic performance.

## 4.2 PART II: TEXTUAL GRADIENT-GUIDED REPAIR MECHANISM

In this section, the intelligent analysis and repair of failure cases using the textual gradient repair mechanism will be used for microscopic and precise logic repair and optimization. We adopt the vanilla prompts for $P_{grad}$ from TextGrad Yuksekgonul et al. (2025). To achieve AutoFloorplan, we design a customized textual loss function $P_{loss}$ and floorplanning algorithm update function $P_{update}$. As shown in Part II of Figure 3, this is the core innovation of the AutoFloorplan framework, which is the key to differentiate it from all the traditional discard-restart models. It perfectly responds to the aforementioned challenge of invalid individuals being wasted, and we think that every invalid individual generation contains a valuable learning signal. When code evaluators discovers that an LLM-generated algorithm has produced an invalid floorplans, the system deeply analyzes the root cause of the algorithm's failure and translates it into structured, instructive corrective feedback. This feedback is cleverly compared to the gradient in text space.

### 4.2.1 DEFINE VARIABLE.

The  of the error and the specific cause of the <Error> (e.g., Module A overlaps with Module B) are packaged and defined as a variable $x$ to be optimized. We define the process that produces a response after a prompts action as $\nu \leftarrow M(x)$ and optimize the variable $x$ by the following steps.

### 4.2.2 CALCULATE LOSS.

Use prompt $P_{loss}$ to instruct the LLM to understand the error, analyze the code and the error message, and locate which part of the algorithm's logic (e.g., perturbation function, positional computation, etc.) is causing the illegal floorplan. This understanding process is the evaluation of the loss. It also provides regression optimization by comparing the results with the current optimal floorplanning.

$$L(x, A_x^{best}) = M(P_{loss}(A_x^{HPWL_{best}}, y_x^{Area_{best}}, Error)) \tag{4}$$

### 4.2.3 COMPUTE GRADIENT.

After that, prompt $P_{grad}$ is used to transform the role of LLM into a feedback engine. Based on its understanding of the error, it generates a piece of targeted corrective feedback in natural language, and updates the textual gradient of each variable in reverse.

$$\frac{\partial L}{\partial x} = M(P_{grad}(L(x, A_x^{best}))) \tag{5}$$

### 4.2.4 OPTIMIZE VARIABLE.

Finally, use prompt $P_{update}$ to instruct LLM to act as an optimizer. It takes the original code and the textual gradients (fix suggestions) and rewrites and fixes the code according to this precise guidance, generating a new, corrected version.

$$x_{new} = M(P_{update}(\frac{\partial L}{\partial x})) \tag{6}$$

The optimization process will loop $num\_iter$ times, during which if the algorithm is repaired, the repair process will end and the algorithm will be included in the elite population, and if the algorithm is still not repaired after reaching the maximum number of loops, the algorithm will be discarded. In summary, the introduction of the textual gradient-guided repair mechanism elevates LLM from a mere generator to a learner capable of self-reflection and repair. Specific prompts are detailed in **Appendix A.3**.

Table 1: Comparison with SOTAs on MCNC and GSRC Benchmarks. Minimize area and HPWL at the same time. The **bold** and underlined numbers indicate the best and second-best performances, respectively. − indicates that optimization is not performed or cannot be measured. (Area Unit: $10^6 \ \mu m^2$, Wire Unit: $10^5 \ \mu m$)

| Circuit | #Macro | #Net | SA | | ICCD'20 | | KDD'22 | | DATE'2024 | | TODAES'2024 | | ICCD'2024 | | Our | |
|---|---|---|---|---|---|---|---|---|---|---|---|---|---|---|---|---|
| | | | Area | Wire | Area | Wire | Area | Wire | Area | Wire | Area | Wire | Area | Wire | Area | Wire |
| **aim33** | 39 | 123 | 1.275 | 0.590 | 1.240 | 0.690 | - | 0.820 | 1.253 | **0.460** | 1.235 | 0.640 | **1.209** | 0.842 | 1.218 | 0.521 |
| **aim49** | 49 | 408 | 39.053 | 14.220 | 38.650 | 17.240 | - | 13.750 | 38.127 | 9.740 | 38.028 | 13.250 | **37.237** | 13.133 | 40.340 | **6.984** |
| **n10** | 10 | 118 | 0.238 | 0.180 | 0.239 | 0.170 | - | 0.410 | - | - | **0.234** | **0.140** | 0.313 | 0.408 | 0.237 | 0.182 |
| **n30** | 30 | 349 | 0.228 | 0.480 | 0.223 | 0.490 | - | 1.120 | - | - | **0.218** | 0.460 | 0.308 | 1.035 | 0.220 | **0.448** |
| **n50** | 50 | 485 | 0.221 | 0.990 | 0.215 | 1.020 | - | 1.630 | - | - | **0.211** | 0.950 | 0.221 | 1.505 | 0.226 | **0.740** |
| **n100** | 100 | 576 | 0.205 | 1.540 | 0.195 | 1.550 | - | 3.370 | 0.192 | 1.250 | 0.190 | 1.370 | **0.186** | 2.951 | 0.209 | **1.148** |
| **n200** | 200 | 1274 | 0.207 | 3.340 | 0.215 | 3.480 | - | 3.520 | 0.200 | **3.100** | 0.197 | 3.260 | 0.193 | 3.229 | **0.193** | 3.332 |
| **n300** | 300 | 1632 | 0.329 | 5.440 | 0.340 | 5.250 | - | 4.770 | 0.309 | 4.770 | 0.301 | 4.860 | 0.315 | 4.281 | **0.292** | **3.074** |

## 5 EXPERIMENTS

### 5.1 EXPERIMENTAL SETTINGS

#### 5.1.1 DATASETS.

We make extensive use of the two public datasets in floorplanning, respectively. GSRC[1] contains six circuits where the number of blocks varies from 10 to 300, while MCNC[2] contains two circuits named ami33 and ami49. A brief summary of these circuits is shown in **Appendix A.4**. Note that the largest circuit in the MCNC, n300, is significantly larger than most industrial circuits, as described in Mallappa et al. (2024); He et al. (2020).

#### 5.1.2 BASELINES.

We evaluate the performance of AutoFloorplan by comparing with three classic baselines including SA-based methods, ICCD'20 He et al. (2020) as well as KDD'22 Amini et al. (2022), and three SOTA methods, DATE'2024 Yang et al. (2024a), TODAES'2024 Yang et al. (2024b) and ICCD'2024 Guan et al. (2024). Details about baselines can be found in **Appendix A.5**.

#### 5.1.3 REPRODUCIBILITY.

For AutoFloorplan, the number of evolutionary learning iterative search generations is set to 20, the textual gradient-guided repair process is set to 2, and four elite individuals are randomly selected for Action e1 and e2. All experiments are completed using GPT-4o by default unless otherwise stated. All search experiments are conducted on a machine equipped with four NVIDIA GeForce RTX 3090 GPUs, two Intel(R) Xeon(R) Silver 4210 CPUs (2.20 GHz), and 252GB of RAM.

### 5.2 COMPARISON WITH BASELINES

To comprehensively evaluate the performance of the AutoFloorplan framework, we conduct extensive experiments on eight widely used public benchmark circuits. Our proposed approach is compared in depth with a variety of state-of-the-art SOTA floorplanning algorithms in terms of key metrics such as area optimization, wire length minimization, and computation time.

Table 1 shows in detail the results of AutoFloorplan compared to various SOTA algorithms for the simultaneous optimization of both area and wire length objectives. Across the eight benchmark circuits, AutoFloorplan achieves the best currently known results (marked in **bold**) on area for two circuits (n200, n300) and on wire length for five circuits (aim49, n30, n50, n100, n300). Notably, on the most challenging large-scale circuits (n200 and n300), AutoFloorplan outperforms all previous SOTA methods across the board. For example, on the n300 circuit, AutoFloorplan achieves a significant 10.3% improvement in area (4.281) over the next best TODAES'2024 (4.770), and a dramatic 28.4% reduction in wire length (3.074) over the next best ICCD'2024 (4.292). In some used cases, AutoFloorplan demonstrated its ability to find high-quality pareto-optimal solutions, even when it

---

[1] http://vlsicad.eecs.umich.edu/BK/GSRCbench/
[2] http://vlsicad.eecs.umich.edu/BK/MCNCbench/

Table 2: Wirelength Minimization on MCNC and GSRC Benchmarks(Wire Unit: $10^5$ $\mu m$)

| Circuit | #Blocks | #Net | SA | ICCD'20 | TODAES'24 | Our |
|---------|---------|------|------|---------|-----------|------|
| **ami33** | 33 | 123 | 0.39 | 0.40 | **0.36** | 0.39 |
| **ami49** | 49 | 408 | 7.12 | 7.33 | **7.03** | 7.78 |
| **n100** | 100 | 576 | 1.22 | 1.25 | 1.09 | **1.04** |
| **n200** | 200 | 1274 | 2.96 | 2.99 | 2.91 | **2.17** |
| **n300** | 300 | 1632 | 4.63 | 4.54 | 4.47 | **3.84** |

Table 3: The Comparison of Runtime. (Unit: s)

| Circuit | SA | ICCD'20 | DATE'24 | TODAES'24 | ICCD'24 | Our |
|---------|------|---------|---------|-----------|---------|------|
| **ami33** | 82 | 43 | 88 | 102 | 41 | 976 |
| **ami49** | 165 | 67 | 202 | 242 | 49 | 896 |
| **n10** | 32 | 18 | - | 30 | 22 | 4 |
| **n30** | 67 | 52 | - | 74 | 40 | 600 |
| **n50** | 132 | 89 | - | 301 | 59 | 58 |
| **n100** | 396 | 389 | 1425 | 1513 | 109 | 51 |
| **n200** | 1102 | 785 | 3603 | 3875 | 292 | 2790 |
| **n300** | 2032 | 3767 | 8031 | 8322 | 542 | 2654 |
| **Average** | 501 | 651 | - | 1807 | 144 | 1003 |

failed to achieve first place in both metrics. For example, on the ami33 circuit, AutoFloorplan's wire length (0.521) is the best among all methods, and its area (1.218) is also at a sub-optimal level. This demonstrates that the heuristics generated by LLM are highly flexible and are able to explore high-quality solutions with different optimization focuses that are hard to reach by traditional algorithms. Overall, AutoFloorplan obtained 7 optimal and 4 sub-optimal metrics out of 8 circuits, totaling 11 winning records, which is far more than any other single algorithm. This is a strong testament to the robustness of the AutoFloorplan framework and the general effectiveness of the heuristics it evolves. Specific foorplanning heuristic algorithm details can be found in the **Appendix A.6**. Visualizations of the floorplans generated by the corresponding heuristic floorplanning algorithm are shown in **Appendix A.8**.

In addition, we provide case studies in **Appendix A.7** designed to demonstrate the practical application and effectiveness of the AutoFloorplan optimization process, including the raw algorithms and code, violated constraint error, textual loss, textual gradients, textual updates, and textual gradient-guided repair algorithms and code.

To further validate the performance of AutoFloorplan under a specific optimization objective, we place it in a scenario where only wire length minimization is the objective and compare it with the SOTA method. The results are shown in Table 2. The performance of AutoFloorplan in this special evaluation is equally impressive, especially when dealing with large-scale problems. AutoFloorplan achieves the best wire length results (2.17 and 3.84, respectively) on both the largest and most complex n200 and n300 circuits, outperforming all the compared methods. This indicates that the heuristic algorithm evolved by AutoFloorplan has better scalability and its sophisticated decision logic can more effectively handle the challenges posed by the dramatic increase in the number of modules and network connections. Although on some small and medium-sized circuits (e.g., ami49), AutoFloorplan's result (7.78) is slightly inferior to TODAES'24 (7.03), the difference is very small. This suggests that AutoFloorplan is not a specialized algorithm for a particular size of problem, but rather a general framework that is competitive across the board.

Table 3 compares the running times of different algorithms. Efficient solution speed is crucial for the usefulness of automated design tools. In terms of average runtime, AutoFloorplan (1003) is not the fastest algorithm, but its efficiency is within acceptable limits and significantly faster than the SOTA algorithm TODAES'24 (1807). It is important to emphasize that the running time of AutoFloorplan is mainly consumed in the evolutionary discovery phase of the heuristic algorithm. This is a one-time, offline training cost. Once a high-performance heuristic has been evolved, it can be quickly applied to solve similar or new floorplanning problems. Combined with the results in table 1, AutoFloorplan achieves an excellent mass-to-time performance ratio. For example, while ICCD'24 is faster on the n300 (542 vs. 2654), AutoFloorplan achieves a performance improvement of up to 10.3% in area and 28.4% in wire length. In a chip design that seeks the ultimate PPA, this trade-off of a reasonable time cost for a huge performance gain is very valuable.

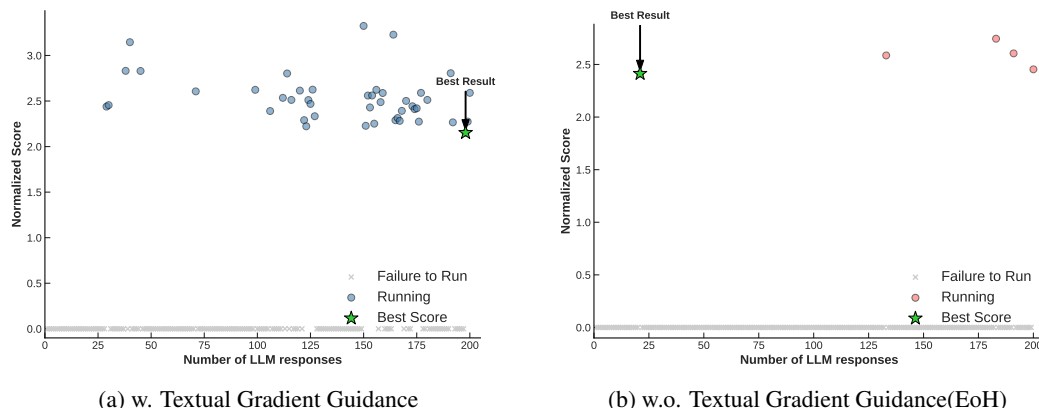

(a) w. Textual Gradient Guidance

(b) w.o. Textual Gradient Guidance(EoH)

Figure 4: The ablation experiments with the textual gradient repair module (w. then the module is retained, w.o. then the module is removed).

## 5.3 ABLATION STUDY

In this section, we make an ablation study on the ablation experiment on the textual gradient repair module, as shown in Figure 4, w. Textual Gradient Guidance denotes the experiment of retaining the module, and w.o. Textual Gradient Guidance denotes the removal of the textual gradient repair module, i.e., retaining the evolutionary search process, which at this point can be approximated as EoH Liu et al. (2024), on the circuit ami33, limiting the number of responses of LLM to 200, it can be seen that under w. Textual Gradient Guidance condition, there are more legitimate individuals, and the effective response rate of LLM is 22%, compared with that of w.o. Textual Gradient Guidance condition, which is only 2.5%, then it will greatly slow down the iterative optimization process of the algorithm.

Table 4: AutoFloorplan $P_{loss}$ ablation experiment (w. then the module is retained, w.o. then the module is removed).

| Model | Variant | HPWL | Area | Score | EEP |
|---|---|---|---|---|---|
| Gpt 4o-mini | w. $P_{loss}$ | 0.485 | 1.317 | 2.151 | 22.00% |
| | w.o. $P_{loss}$ | 0.691 | 1.324 | 2.610 | 3% |

Regarding Prompts within the textual gradient optimization framework, as shown in Appendix A.3, except for $P_{loss}$, $P_{grad}$ and $P_{update}$ are indispensable of the textual gradient optimization framework. To evaluate the sensitivity of $P_{loss}$, we intentionally replaced the task-specific loss description with an extremely generic prompt — "Help me optimize the algorithm." As shown in the ablation Table 4, this simplification leads to significantly degraded performance, with HPWL, Area, and Score all worsening and EEP dropping dramatically.

In summary, traditional evolutionary frameworks usually adopt a simple discard-restart strategy when confronted with invalid individuals generated by LLM, which leads to a large amount of wasted computational resources and inefficient exploration into a valuable learning opportunity. By analyzing the causes of constraint violations and generating structured corrective feedback (i.e., textual gradients), AutoFloorplan can guide LLMs to targeted logic fixes. This correct rather than discard strategy greatly improves the discovery of legitimate and high-performance heuristics.

## 5.4 PARAMETER ANALYSIS

This section explores the impact of various components and hyperparameters on AutoFloorplan.

### 5.4.1 NUMBER OF TEXTUAL GRADIENT-GUIDED REPAIR.

The first is about the analysis of the hyperparameter $num\_iter$, we set the evolutionary learning iterative search generation fixed to 5, and then change the size of $num\_iter$. The results are shown

Table 5: Parametric analysis experiments on the number of textual gradient-guided repairs performed on circuit ami33. (Area Unit: $10^6 \ \mu m^2$, Wire Unit: $10^5 \ \mu m$)

| $num\_iter$ | HPWL | Area | Score | EEP |
|---|---|---|---|---|
| 2 | 0.485 | 1.317 | 2.151 | 22.00% |
| 3 | 0.347 | 1.357 | 1.886 | 19.50% |
| 4 | 0.501 | 1.295 | 2.168 | 12.36% |

in Table 5, where Score is the individual score calculated according to formula 3, Effective Effect Probability(EEP) indicates the effective response rate of LLMs, which can be used to measure the efficiency of AutoFloorplan's work. It can be seen that simply increasing $num\_iter$ does not increase the EEP, because some individual deviations are too large to be corrected.

### 5.4.2 CHOICES OF LLMS.

To evaluate the impact of different LLMs on AutoFloorplan, we replace the original gpt-4o-mini with DeepSeek-R1 and Gemini-2.5-Pro, respectively. The corresponding results are presented in Table 6. As shown, the enhanced modeling capabilities of these more advanced LLMs lead to notable improvements in both floorplanning quality and estimated performance efficiency (EPE). These findings highlight the scalability of AutoFloorplan and suggest that its performance can continue to improve as LLM technology advances.

Table 6: Analytical experiments with different LLMs types on circuit ami33. (Area Unit: $10^6 \ \mu m^2$, Wire Unit: $10^5 \ \mu m$)

| LLM Type | HPWL | Area | Score | EEP |
|---|---|---|---|---|
| Gpt 4o-mini | 0.485 | 1.317 | 2.151 | 22.00% |
| Deepseek-R1 | 0.434 | 1.278 | 2.009 | 29.41% |
| Gemini-2.5-Pro | 0.382 | 1.316 | 1.928 | 42.59% |

## 6 CONCLUSION AND FUTURE WORK

AutoFloorplan proposes a novel evolutionary learning framework for solving NP-hard chip floorplanning problems by automatically discovering heuristics floorplanning algorithm using LLMs. Our innovative textual gradient-guided repair mechanism efficiently transforms invalid heuristics generated by the LLMs into valid ones, thus greatly accelerating the search for high-quality solutions. Extensive experiments on eight public benchmarks show that AutoFloorplan consistently outperforms current SOTA algorithms in terms of area and wire length minimization, achieving excellent results on challenging large-scale circuits. While its evolutionary discovery phase requires an upfront computational cost, its significant performance gains and ability to reduce reliance on human expertise, as well as AutoFloorplan's high degree of parallelism, make AutoFloorplan a powerful and universally effective framework for automated algorithm design in complex optimization problems. In the future, we will advance the application of AutoFloorplan to floorplanning scenarios with more complex constraints.

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

# A APPENDIX

## A.1 SUMMARY OF ALGORITHM ERROR CASES.

| Error Description | Number |
|---|---|
| Blocks beyond the outline | 38 |
| Blocks overlap | 13 |
| Incorrect number of finalized blocks | 5 |
| Python syntax errors | 21 |
| Others | 17 |

Table 7: Individual analysis of error algorithms generated through 100 iterations of LLM continuous running.

We appreciate the reviewer's comment regarding domain-specific requirements in floorplanning. To clarify, we conducted an analysis of over 100 iterations of LLM-generated error algorithms and found that the errors are highly domain-specific:

- 38 cases of "blocks beyond the outline".
- 13 cases of "blocks overlap".
- 5 cases of "incorrect number of finalized blocks".

In addition to Python syntax and other errors. These errors directly reflect structural constraints unique to chip floorplanning rather than generic code-generation issues.

Importantly, simply discarding these error individuals—as would be done in standard LLM code-generation workflows—would drastically shrink the viable population and severely impede both iteration speed and convergence. This is precisely why our method incorporates textual-gradient–guided repair: it enables the algorithmic population to recover from floorplanning-specific constraint violations and continue evolving, which is essential for maintaining diversity and progress.

## A.2 DETAILS OF EVOLUTIONARY PROMPTS.

As shown in the figure below, this paper utilizes the designed six evolutionary hints (i1, m1, m2, e1, e2, d1) to guide the LLMs to explore the floorplanning heuristic algorithm space. The evolutionary prompts (m1, m2) and (e1, e2) function similarly to mutations and crossovers. And prompt d1 models algorithm like DPO, mimicking the contrastive learning approach with positive and negative samples. This enables LLMs to better understand the preferences of different circuit data and floorplanning task.

---

### Prompt for Action *i1*

Please design a floorplanning function to select the optimal position for each block and complete the floorplan for VLSI aiming to minimize HPWL and area.The required optimized results are HPWL: **<Target HPWL Result >** and Area: **<Target Area Result>**.

First, describe your new algorithm and main steps in one sentence. The description must be inside a brace { }. After the brace containing the description, provide *only* the Python code implementation. The code block must be between <FLOORPLANCODESTART> and <FLOORPLANCODEEND> and contain any necessary `import` statements starting directly, followed by the function definition `def floorplan(...)`. The function implementation must contain only valid Python code. Do not include the algorithm description or any other explanatory text within the code block itself.
The function floorplan should accept 3 input(s): 'blocks', 'max_chip_size', 'nets'. The function should return 1 output(s): 'placements'.
Input:
- blocks: A list of blocks. Blocks are represented as a list [[block_id1, initial_x, initial_y, width, height], [block_id2, initial_x, initial_y, width, height], ...], [list(b) for b in blocks]. All values are integers. initial_x and initial_y might be placeholders (e.g., 0) if no initial placement is given.
- max_chip_size: A list representing the chip's maximum boundary dimensions [max_width, max_height]. Both values are integers.
- nets: A list of nets. Each net is a list containing the block_ids connected by that net.

Output:
- placements: A list of blocks, where each list represents the final placement of a block in the format [[block_id1, final_x, final_y, width, height], [block_id2, final_x, final_y, width, height], ...], [list(b) for b in blocks]. All values must be integers. The list should contain placements for all input blocks.

Constraints and Objectives:
1. No Overlap: Placed blocks must not overlap each other. The overlap area between any two distinct blocks must be zero.
2. Boundary Adherence: All placed blocks must be entirely within the chip maximum boundaries defined by chip_size (i.e., 0 <= x < chip_width and 0 <= y < chip_height for the bottom-left corner, and x + width <= chip_width and y + height <= chip_height for the top-right corner).
3. Minimize HPWL: The primary objective is to minimize the total Half-Perimeter Wire Length (HPWL) calculated based on the final placement and the provided nets.
4. Minimize area: The main goal is to minimize the area occupied by the final layouts of blocks with the formula area=(max(blocks(x))-min(blocks(x)))*(max(blocks(y))-min(blocks(y))).

Do not give any additional explanations outside the initial description brace and the final code block.

---

Figure 5: The specific details of prompt *i1*.

## Prompt for Action *m1*

Please design a floorplanning function to select the optimal position for each block and complete the floorplan for VLSI aiming to minimize HPWL and area.The required optimized results are HPWL: **<Target HPWL Result>** and Area: **<Target Area Result>**.

I have one algorithm with its code as follows.
**Algorithm description: <Algorithm Think> Code:<Algorithm Code> HPWL:<HPWL Result> A rea:<Area Result>**
Please introducing the code modules or modifying the core mechanism to enhance its exploration capabilities based on the existing floorplan heuristic algorithm.
First, describe your new algorithm and main steps in one sentence. The description must be inside a brace. Next, implement it in Python as a function named floorplan. The code block must be between <FLOORPLANCODESTART> and <FLOORPLANCODEEND> and contain any necessary `import` statements starting directly.This function should accept 3 input(s): 'blocks', 'max_chip_size', 'nets'. The function should return 1 output(s): 'placements'.
Input:
- blocks: A list of blocks. Blocks are represented as a list [[block_id1, initial_x, initial_y, width, height], [block_id2, initial_x, initial_y, width, height], ...], [list(b) for b in blocks]. All values are integers. initial_x and initial_y might be placeholders (e.g., 0) if no initial placement is given.
- max_chip_size: A list representing the chip's maximum boundary dimensions [max_width, max_height]. Both values are integers.
- nets: A list of nets. Each net is a list containing the block_ids connected by that net.

Output:
- placements: A list of blocks, where each list represents the final placement of a block in the format [[block_id1, final_x, final_y, width, height], [block_id2, final_x, final_y, width, height], ...], [list(b) for b in blocks]. All values must be integers. The list should contain placements for all input blocks.

Constraints and Objectives:
1. No Overlap: Placed blocks must not overlap each other. The overlap area between any two distinct blocks must be zero.
2. Boundary Adherence: All placed blocks must be entirely within the chip maximum boundaries defined by chip_size (i.e., 0 <= x < chip_width and 0 <= y < chip_height for the bottom-left corner, and x + width <= chip_width and y + height <= chip_height for the top-right corner).
3. Minimize HPWL: The primary objective is to minimize the total Half-Perimeter Wire Length (HPWL) calculated based on the final placement and the provided nets.
4. Minimize area: The main goal is to minimize the area occupied by the final layouts of blocks with the formula area=(max(blocks(x))-min(blocks(x)))*(max(blocks(y))-min(blocks(y))).

Do not give additional explanations.

Figure 6: The specific details of prompt *m1*.

**Prompt for Action *m2***

Please design a floorplanning function to select the optimal position for each block and complete the floorplan for VLSI aiming to minimize HPWL and area.The required optimized results are HPWL: **<Target HPWL Result>** and Area: **<Target Area Result>**.

I have one algorithm with its code as follows.
**Algorithm description: <Algorithm Think> Code:<Algorithm Code> HPWL:<HPWL Result> Area:<Area Result>**
Without changing the core structure of the existing floorplan heuristic algorithm, performance improvements can only be explored by modifying the parameters.
First, describe your new algorithm and main steps in one sentence. The description must be inside a brace. Next, implement it in Python as a function named floorplan. The code block must be between <FLOORPLANCODESTART> and <FLOORPLANCODEEND> and contain any necessary `import` statements starting directly.This function should accept 3 input(s): 'blocks', 'max_chip_size', 'nets'. The function should return 1 output(s): 'placements'.
Input:
- blocks: A list of blocks. Blocks are represented as a list [[block_id1, initial_x, initial_y, width, height], [block_id2, initial_x, initial_y, width, height], ...], [list(b) for b in blocks]. All values are integers. initial_x and initial_y might be placeholders (e.g., 0) if no initial placement is given.
- max_chip_size: A list representing the chip's maximum boundary dimensions [max_width, max_height]. Both values are integers.
- nets: A list of nets. Each net is a list containing the block_ids connected by that net.

Output:
- placements: A list of blocks, where each list represents the final placement of a block in the format [[block_id1, final_x, final_y, width, height], [block_id2, final_x, final_y, width, height], ...], [list(b) for b in blocks]. All values must be integers. The list should contain placements for all input blocks.

Constraints and Objectives:
1. No Overlap: Placed blocks must not overlap each other. The overlap area between any two distinct blocks must be zero.
2. Boundary Adherence: All placed blocks must be entirely within the chip maximum boundaries defined by chip_size (i.e., 0 <= x < chip_width and 0 <= y < chip_height for the bottom-left corner, and x + width <= chip_width and y + height <= chip_height for the top-right corner).
3. Minimize HPWL: The primary objective is to minimize the total Half-Perimeter Wire Length (HPWL) calculated based on the final placement and the provided nets.
4. Minimize area: The main goal is to minimize the area occupied by the final layouts of blocks with the formula area=(max(blocks(x))-min(blocks(x)))*(max(blocks(y))-min(blocks(y))).

Do not give additional explanations.

Figure 7: The specific details of prompt *m2*.

## Prompt for Action *e1*

Please design a floorplanning function to select the optimal position for each block and complete the floorplan for VLSI aiming to minimize HPWL and area. The required optimized results are HPWL: **<Target HPWL Result>** and Area: **<Target Area Result>**.

I have 4 existing algorithms with their codes as follows:
No.1 algorithm and the corresponding code are:
<Algorithm1 Think>
<Algorithm1 Code>
 HPWL:<HPWL Result> Area:<Area Result>
...
No.4 algorithm and the corresponding code are:
<Algorithm4 Think>
<Algorithm4 Code>
 HPWL:<HPWL Result> Area:<Area Result>
Please help me create a new algorithm that has a totally different form from the given ones.

First, describe your new algorithm and main steps in one sentence. The description must be inside a brace {}. After the brace containing the description, provide *only* the Python code implementation. The code block must be between <FLOORPLANCODESTART> and <FLOORPLANCODEEND> and contain any necessary `import` statements starting directly, followed by the function definition `def floorplan(...)`. The function implementation must contain only valid Python code. Do not include the algorithm description or any other explanatory text within the code block itself.

The function floorplan should accept 3 input(s): 'blocks', 'max_chip_size', 'nets'. The function should return 1 output(s): 'placements'.

Input:
- blocks: A list of blocks. Blocks are represented as a list [[block_id1, initial_x, initial_y, width, height], [block_id2, initial_x, initial_y, width, height], ...], [list(b) for b in blocks]. All values are integers. initial_x and initial_y might be placeholders (e.g., 0) if no initial placement is given.
- max_chip_size: A list representing the chip's maximum boundary dimensions [max_width, max_height]. Both values are integers.
- nets: A list of nets. Each net is a list containing the block_ids connected by that net.

Output:
- placements: A list of blocks, where each list represents the final placement of a block in the format [[block_id1, final_x, final_y, width, height], [block_id2, final_x, final_y, width, height], ...], [list(b) for b in blocks]. All values must be integers. The list should contain placements for all input blocks.

Constraints and Objectives:
1. No Overlap: Placed blocks must not overlap each other. The overlap area between any two distinct blocks must be zero.
2. Boundary Adherence: All placed blocks must be entirely within the chip maximum boundaries defined by chip_size (i.e., 0 <= x < chip_width and 0 <= y < chip_height for the bottom-left corner, and x + width <= chip_width and y + height <= chip_height for the top-right corner).
3. Minimize HPWL: The primary objective is to minimize the total Half-Perimeter Wire Length (HPWL) calculated based on the final placement and the provided nets.
4. Minimize area: The main goal is to minimize the area occupied by the final layouts of blocks with the formula area=(max(blocks(x))-min(blocks(x)))*(max(blocks(y))-min(blocks(y))).

Do not give any additional explanations outside the initial description brace and the final code block.

Figure 8: The specific details of prompt *e1*.

## Prompt for Action *e2*

Please design a floorplanning function to select the optimal position for each block and complete the floorplan for VLSI aiming to minimize HPWL and area. The required optimized results are HPWL: **<Target HPWL Result>** and Area: **<Target Area Result>**.

I have 4 existing algorithms with their codes as follows:
No.1 algorithm and the corresponding code are:
<Algorithm1 Think>
<Algorithm1 Code>
 HPWL:<HPWL Result> Area:<Area Result>
...
No.4 algorithm and the corresponding code are:
<Algorithm4 Think>
<Algorithm4 Code>
 HPWL:<HPWL Result> Area:<Area Result>
Please help me create a new algorithm that is similar in form to the core idea of the given algorithm.

First, describe your new algorithm and main steps in one sentence. The description must be inside a brace {}. After the brace containing the description, provide *only* the Python code implementation. The code block must be between <FLOORPLANCODESTART> and <FLOORPLANCODEEND> and contain any necessary `import` statements starting directly, followed by the function definition `def floorplan(...)`. The function implementation must contain only valid Python code. Do not include the algorithm description or any other explanatory text within the code block itself.
The function floorplan should accept 3 input(s): 'blocks', 'max_chip_size', 'nets'. The function should return 1 output(s): 'placements'.

Input:
- blocks: A list of blocks. Blocks are represented as a list [[block_id1, initial_x, initial_y, width, height], [block_id2, initial_x, initial_y, width, height], ...], [list(b) for b in blocks]. All values are integers. initial_x and initial_y might be placeholders (e.g., 0) if no initial placement is given.
- max_chip_size: A list representing the chip's maximum boundary dimensions [max_width, max_height]. Both values are integers.
- nets: A list of nets. Each net is a list containing the block_ids connected by that net.

Output:
- placements: A list of blocks, where each list represents the final placement of a block in the format [[block_id1, final_x, final_y, width, height], [block_id2, final_x, final_y, width, height], ...], [list(b) for b in blocks]. All values must be integers. The list should contain placements for all input blocks.

Constraints and Objectives:
1. No Overlap: Placed blocks must not overlap each other. The overlap area between any two distinct blocks must be zero.
2. Boundary Adherence: All placed blocks must be entirely within the chip maximum boundaries defined by chip_size (i.e., 0 <= x < chip_width and 0 <= y < chip_height for the bottom-left corner, and x + width <= chip_width and y + height <= chip_height for the top-right corner).
3. Minimize HPWL: The primary objective is to minimize the total Half-Perimeter Wire Length (HPWL) calculated based on the final placement and the provided nets.
4. Minimize area: The main goal is to minimize the area occupied by the final layouts of blocks with the formula area=(max(blocks(x))-min(blocks(x)))*(max(blocks(y))-min(blocks(y))).

Do not give any additional explanations outside the initial description brace and the final code block.

Figure 9: The specific details of prompt *e2*.

## Prompt for Action *d1*

Please design a floorplanning function to select the optimal position for each block and complete the floorplan for VLSI aiming to minimize HPWL and area. The required optimized results are HPWL: **<Target HPWL Result>** and Area: **<Target Area Result>**.

I have 2 existing algorithms with their codes as follows:
No.1 algorithm and the corresponding code are:
<Algorithm1 Think>
<Algorithm1 Code>
 HPWL:<HPWL Result> Area:<Area Result>
No.2 algorithm and the corresponding code are:
<Algorithm2 Think>
<Algorithm2 Code>
 HPWL:<HPWL Result> Area:<Area Result>
First, analyze the principles of both. Second, by comparing their scores, evaluate why the filter with the larger of the scores is more suitable for the graphs. Finally, help me to create a new spectral graph neural network layer, and describe your new algorithm and main steps in one sentence.
The description must be inside a brace {}. After the brace containing the description, provide *only* the Python code implementation. The code block must be between <FLOORPLANCODESTART> and <FLOORPLANCODEEND> and contain any necessary `import` statements starting directly, followed by the function definition `def floorplan(...)`. The function implementation must contain only valid Python code. Do not include the algorithm description or any other explanatory text within the code block itself.
The function floorplan should accept 3 input(s): 'blocks', 'max_chip_size', 'nets'. The function should return 1 output(s): 'placements'.
Input:
- blocks: A list of blocks. Blocks are represented as a list [[block_id1, initial_x, initial_y, width, height], [block_id2, initial_x, initial_y, width, height], ...], [list(b) for b in blocks]. All values are integers. initial_x and initial_y might be placeholders (e.g., 0) if no initial placement is given.
- max_chip_size: A list representing the chip's maximum boundary dimensions [max_width, max_height]. Both values are integers.
- nets: A list of nets. Each net is a list containing the block_ids connected by that net.

Output:
- placements: A list of blocks, where each list represents the final placement of a block in the format [[block_id1, final_x, final_y, width, height], [block_id2, final_x, final_y, width, height], ...], [list(b) for b in blocks]. All values must be integers. The list should contain placements for all input blocks.

Constraints and Objectives:
1. No Overlap: Placed blocks must not overlap each other. The overlap area between any two distinct blocks must be zero.
2. Boundary Adherence: All placed blocks must be entirely within the chip maximum boundaries defined by chip_size (i.e., 0 <= x < chip_width and 0 <= y < chip_height for the bottom-left corner, and x + width <= chip_width and y + height <= chip_height for the top-right corner).
3. Minimize HPWL: The primary objective is to minimize the total Half-Perimeter Wire Length (HPWL) calculated based on the final placement and the provided nets.
4. Minimize area: The main goal is to minimize the area occupied by the final layouts of blocks with the formula area=(max(blocks(x))-min(blocks(x)))*(max(blocks(y))-min(blocks(y))).

Do not give any additional explanations outside the initial description brace and the final code block.

Figure 10: The specific details of prompt *d1*.

## A.3 DETAILS OF TEXTUAL GRADIENT-GUIDED REPAIR PROMPTS.

We adopt the vanilla prompts for $P_{grad}$ from TextGrad (Yuksekgonul et al. 2025). To achieve AutoFloorplan, we design a customized textual loss function $P_{loss}$ and floorplanning algorithm update function $P_{update}$, as listed in the follow figures:

---

### Prompt for Loss Calculation $P_{Loss}$

You should evaluate the floorplanning algorithm generated by the model.

**User Query**.
The desired goal is: HPWL is {Target HPWL Result}, area is {Target Area Result}.

Below is an algorithm for floor planning which has some errors, please analyze the reasons for the errors and give modifications to make the code work properly and optimize HPWL and Area to the maximum.

**Error**:{Error}

**Do NOT generate a response to the query. be concise.**

---

Figure 11: The specific details of prompt $P_{loss}$.

---

### Prompt for Compute Gradient $P_{grad}$

You are part of an optimization system that improves text (i.e., variable). You will be asked to creatively and critically improve prompts, solutions to problems, code, or any other text-based variable. You will receive some feedback, and use the feedback to improve the variable. The feedback may be noisy, identify what is important and what is correct. Pay attention to the role description of the variable, and the context in which it is used. This is very important: You MUST give your response by sending the improved variable between {new_variable_start_tag} {{improved variable}} {new_variable_end_tag} tags. The text you send between the tags will directly replace the variable.

### Glossary of tags that will be sent to you:
# - <LM_SYSTEM_PROMPT>: The system prompt for the language model.
# - <LM_INPUT>: The input to the language model.
# - <LM_OUTPUT>: The output of the language model.
# - <FEEDBACK>: The feedback to the variable.
# - <CONVERSATION>: The conversation history.
# - <FOCUS>: The focus of the optimization.
# - <ROLE>: The role description of the variable.'

---

Figure 12: The specific details of prompt $P_{grad}$.

---

### Prompt for Variable Update $P_{update}$

Given a text variable and its textual gradient (modification suggestion), please optimize the algorithm and code in combination with the gradient.

**Gradient** <Algorithm code textual gradient>

**The Algorithm and Code** <Algorithm Code>

**Other Information** First, describe your new algorithm and main steps in one sentence. The description must be inside a brace {}. After the brace containing the description, provide *only* the Python code implementation. The code block must be between <FLOORPLANCODESTART> and <FLOORPLANCODEEND> and contain any necessary `import` statements starting directly, followed by the function definition `def floorplan(...)` . The function implementation must contain only valid Python code. Do not include the algorithm description or any other explanatory text within the code block itself.

The function Please design a floorplanning function to select the optimal position for each block and complete the floorplan for VLSI aiming to minimize HPWL and area. should accept 3 input(s): 'blocks', 'max_chip_size', 'nets'. The function should return 1 output(s): 'placements'.

Input:
- blocks: A list of blocks. Blocks are represented as a list [[block_id1, initial_x, initial_y, width, height], [block_id2, initial_x, initial_y, width, height], ...], [list(b) for b in blocks]. All values are integers. initial_x and initial_y might be placeholders (e.g., 0) if no initial placement is given.
- max_chip_size: A list representing the chip's maximum boundary dimensions [max_width, max_height]. Both values are integers.
- nets: A list of nets. Each net is a list containing the block_ids connected by that net.

Output:
- placements: A list of blocks, where each list represents the final placement of a block in the format [[block_id1, final_x, final_y, width, height], [block_id2, final_x, final_y, width, height], ...], [list(b) for b in blocks]. All values must be integers. The list should contain placements for all input blocks.

Constraints and Objectives:
1. No Overlap: Placed blocks must not overlap each other. The overlap area between any two distinct blocks must be zero.
2. Boundary Adherence: All placed blocks must be entirely within the chip maximum boundaries defined by chip_size (i.e., 0 <= x < chip_width and 0 <= y < chip_height for the bottom-left corner, and x + width <= chip_width and y + height <= chip_height for the top-right corner).
3. Minimize HPWL: The primary objective is to minimize the total Half-Perimeter Wire Length (HPWL) calculated based on the final placement and the provided nets.
4. Minimize area: The main goal is to minimize the area occupied by the final layouts of blocks with the formula area=(max(blocks(x))-min(blocks(x)))*(max(blocks(y))-min(blocks(y))).

Do not give any additional explanations outside the initial description brace and the final code block.

Figure 13: The specific details of prompt $P_{update}$.

## A.4 DETAILS OF DATASETS.

Table 8 presents the statistical data for all benchmark datasets used in this work.

Table 8: Benchmark Information.

| Circuit | #Blocks | #I/O | #Net | #Outline |
|---------|---------|------|------|----------|
| ami33 | 33 | 42 | 123 | (1326,1205) |
| ami49 | 49 | 22 | 408 | (5336,7673) |
| n10 | 10 | 69 | 118 | (800,800) |
| n30 | 30 | 212 | 349 | (800,800) |
| n50 | 50 | 209 | 485 | (800,800) |
| n100 | 100 | 334 | 885 | (800,800) |
| n200 | 200 | 564 | 1585 | (800,800) |
| n300 | 300 | 569 | 1893 | (800,800) |

### A.5 DETAILS OF BASELINES.

To provide a comprehensive evaluation of our proposed methodology, we compare it with a series of baseline approaches that represent the current state-of-the-art in chip floorplanning. These include traditional heuristic-based algorithms and modern reinforcement learning (RL)-driven frameworks, detailed as follows:

#### A.5.1 SIMULATED ANNEALING (SA).

Simulated Annealing is a classical metaheuristic approach applied to floorplanning, relying on Sequence Pair (SP) representation to encode geometric relationships among macros. Optimization is achieved by introducing controlled perturbations and gradually cooling the system based on a temperature schedule. Although capable of escaping local minima to some extent, SA often struggles with complex designs and may get trapped in suboptimal configurations. In our evaluation, we tuned SA hyperparameters (initial temperature: $10^6$ to $10^8$, cooling rate: 0.97, iterations: 50–200, termination temperature: $10^{-11}$) and report the best-performing results. The neighborhood function follows TODAES'24 (Yang et al. 2024b).

#### A.5.2 ICCD'2020 (HE ET AL. 2020B).

RL-based Automatic Heuristic Design This work pioneers the use of reinforcement learning for automatic heuristic discovery. Instead of directly constructing solutions, an RL agent is trained to perform local search by selecting neighboring configurations within the Sequence Pair space. The learned policy enables the agent to effectively navigate the solution space, demonstrating notable improvements over manually crafted heuristics.

#### A.5.3 KDD'22 (AMINI ET AL. 2022).

End-to-End RL with Hypergraph Neural Networks KDD'22 introduces a fully end-to-end reinforcement learning framework for floorplanning. The RL agent predicts both the block ID and its placement location in a single step. A hypergraph neural network encodes the circuit netlist, while a Transformer-based action selection mechanism handles generalization across netlists of varying sizes. The framework exhibits strong transferability, maintaining constant model parameters regardless of circuit scale, and achieves state-of-the-art results on GSRC and MCNC benchmarks.

#### A.5.4 TODEAS'24 (YANG ET AL. 2024B).

EAGAT + Transformer-Based RL Framework TODAES'24 proposes an end-to-end RL method incorporating Edge-Aware Graph Attention Networks (EAGAT) and a hierarchical decoder based on Transformers and attention pointers. Hindsight experience replay further boosts sample efficiency. The agent efficiently captures macro connectivity and layout structure, allowing learned policies to generalize to unseen netlists. Experimental results show significant area and wirelength reductions on public benchmarks.

#### A.5.5 DATE'24 (YANG ET AL. 2024A).

EGAT + Supervised Learning Hybrid DATE'24 formulates floorplanning as a Markov Decision Process and introduces an end-to-end RL framework using an Edge-Augmented Graph Attention Network (EGAT), MLPs, and gated self-attention. The framework integrates dense rewards and a multi-action mechanism. Additionally, it incorporates supervised pretraining on synthetic netlists to leverage prior knowledge. Results reveal considerable gains in area and wirelength minimization versus prior methods.

#### A.5.6 ICCD'24 (GUAN ET AL. 2024).

HyperGCN + Deep Transformer Q-Network (DTQN) ICCD'24 addresses the shortcomings of simplified RL models by enhancing the MDP formulation with sophisticated state representation. HyperGCN captures hypergraph-encoded netlist features, while a Transformer-based Deep Q-Network processes long-range interdependencies among modules. The Transformer encoder uses multi-head

self-attention to account for global design dynamics, leading to superior performance in placement-aware optimization. The method achieves leading performance and marks the first known application of Transformers in floorplanning.

## A.6 SPECIFIC FLOORPLANNING ALGORITHMIC DETAILS.

It is evident that AutoFloorplan generates heuristic algorithms with distinct approaches tailored to the specific characteristics of different chips. Below, we will analyze each generated algorithm to determine whether the rules discovered by AutoFloorplan are interpretable or consistent with human design intuition.

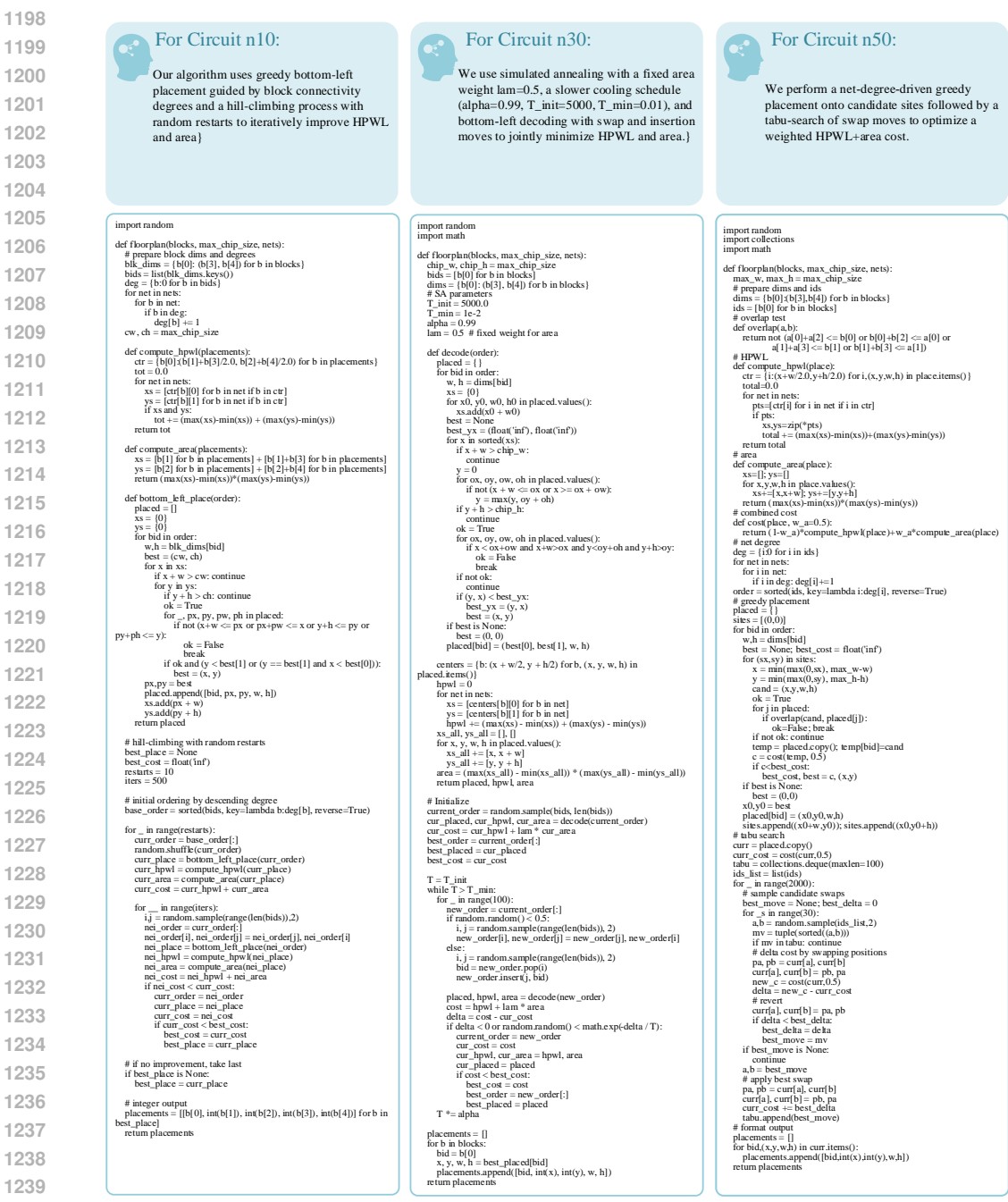

Figure 14: Specific details of the floorplanning heuristic algorithm designed for circuit n10, n30 and n50.

**For Circuit n100:**

Use simulated annealing with a modified cost function cost = 0.8*HPWL + 0.0008*area, an increased initial temperature, slower cooling, and swap/shift moves to optimize block placement.

```python
import random
import math
import copy

def floorplan(blocks, max_chip_size, nets):
    chip_w, chip_h = max_chip_size
    # block dims
    blk_info = {b[0]:(b[3],b[4]) for b in blocks}
    blk_ids = [b[0] for b in blocks]

    # compute HPWL
    def hpwl(pl):
        centers = {b:(x + w/2.0, y + h/2.0) for b,(x,y,w,h) in pl.items()}
        total = 0.0
        for net in nets:
            xs = [centers[b][0] for b in net if b in centers]
            ys = [centers[b][1] for b in net if b in centers]
            if xs and ys:
                total += (max(xs)-min(xs)) + (max(ys)-min(ys))
        return total

    # compute bounding area
    def area(pl):
        xs = [x for x,y,w,h in pl.values()] + [x+w for (x,y,w,h) in
pl.values()]
        ys = [y for x,y,w,h in pl.values()] + [y+h for (x,y,w,h) in
pl.values()]
        return (max(xs)-min(xs)) * (max(ys)-min(ys))

    # check legality
    def valid(pl):
        items = list(pl.items())
        for b,(x,y,w,h) in items:
            if x<0 or y<0 or x+w>chip_w or y+h>chip_h:
                return False
            for i in range(len(items)):
                _,(x1,y1,w1,h1) = items[i]
                for j in range(i+1,len(items)):
                    _,(x2,y2,w2,h2) = items[j]
                    if not (x1+w1<=x2 or x2+w2<=x1 or y1+h1<=y2 or
y2+h2<=y1):
                        return False
        return True

    # initial next-fit row
    placement = {}
    rx = ry = rh = 0
    for b in sorted(blk_ids, key=lambda b: blk_info[b][1],
reverse=True):
        w,h = blk_info[b]
        if rx + w > chip_w:
            ry += rh; rx = 0; rh = 0
        if ry + h > chip_h:
            ry = 0
        placement[b] = (rx, ry, w, h)
        rh = max(rh, h); rx += w

    # SA params
    w_hpwl = 0.8
    w_area = 0.0008
    T = 5000.0
    cooling = 0.99
    iters = 20000
    swap_p = 0.5

    cur = copy.deepcopy(placement)
    cur_cost = w_hpwl * hpwl(cur) + w_area * area(cur)
    best_pl = copy.deepcopy(cur)
    best_cost = cur_cost

    for _ in range(iters):
        new = copy.deepcopy(cur)
        if random.random()< swap_p:
            b1,b2 = random.sample(blk_ids,2)
            new[b1],new[b2] = new[b2],new[b1]
        else:
            b = random.choice(blk_ids)
            x,y,w,h = new[b]
            dx = random.randint(-chip_w//30, chip_w//30)
            dy = random.randint(-chip_h//30, chip_h//30)
            nx = max(0, min(chip_w-w, x+dx))
            ny = max(0, min(chip_h-h, y+dy))
            new[b] = (nx, ny, w, h)

        if not valid(new):
            T *= cooling
            continue

        new_cost = w_hpwl * hpwl(new) + w_area * area(new)
        dE = new_cost - cur_cost
        if dE < 0 or random.random()< math.exp(-dE/T):
            cur, cur_cost = new, new_cost
            if new_cost < best_cost:
                best_cost, best_pl = new_cost, copy.deepcopy(new)
        T *= cooling

    # output placements
    placements = []
    for b in blk_ids:
        x,y,w,h = best_pl[b]
        placements.append([b, int(x), int(y), int(w), int(h)])
    return placements
```

**For Circuit n200:**

A hybrid simulated-annealing algorithm employing both random block-swap and directional slide moves with a dynamic cost weight that shifts emphasis from HPWL to area.

```python
import random
import math
import copy

def floorplan(blocks, max_chip_size, nets):
    max_w, max_h = max_chip_size
    block_ids = [b[0] for b in blocks]
    size_map = {b[0]:(b[3], b[4]) for b in blocks}

    # Greedy initial row-based packing
    def initial_pack():
        placements = {}
        cur_x = 0; cur_y = 0; row_h = 0
        for bid in sorted(block_ids, key=lambda b: size_map[b][1],
reverse=True):
            w,h = size_map[bid]
            if cur_x + w > max_w:
                cur_x = 0
                cur_y += row_h
                row_h = 0
            if cur_y + h > max_h:
                cur_y = 0
            placements[bid] = [cur_x, cur_y, w, h]
            row_h = max(row_h, h)
            cur_x += w
        return placements

    # HPWL
    def calc_hpwl(pl):
        hp = 0
        for net in nets:
            xs = []; ys = []
            for bid in net:
                x,y,w,h = pl[bid]
                xs.append(x + w/2); ys.append(y + h/2)
            hp += (max(xs)-min(xs)) + (max(ys)-min(ys))
        return hp

    # Area
    def calc_area(pl):
        xs = []; ys = []
        for bid,(x,y,w,h) in pl.items():
            xs += [x, x+w]; ys += [y, y+h]
        return (max(xs)-min(xs)) * (max(ys)-min(ys))

    # Validity
    def valid(pl):
        for bid,(x,y,w,h) in pl.items():
            if x<0 or y<0 or x+w>max_w or y+h>max_h:
                return False
        items = list(pl.items())
        for i in range(len(items)):
            b1,(x1,y1,w1,h1) = items[i]
            for j in range(i+1,len(items)):
                b2,(x2,y2,w2,h2) = items[j]
                if x1 < x2+w2 and x2 < x1+w1 and y1 < y2+h2 and y2 <
y1+h1:
                    return False
        return True

    # Simulated annealing with dynamic alpha
    T_init = 10000.0
    T = T_init
    T_min = 1e-3
    decay = 0.995
    inner = 200
    alpha_start = 0.2
    alpha_end = 0.01

    def cost(pl, alpha):
        return calc_hpwl(pl) + alpha * calc_area(pl)

    # Initialize
    best_pl = initial_pack()
    if not valid(best_pl):
        raise ValueError("Initial packing invalid")
    best_cost = cost(best_pl, alpha_start)
    cur_pl = copy.deepcopy(best_pl)
    cur_cost = best_cost

    # SA loop
    while T > T_min:
        # dynamic alpha
        frac = (T / T_init)
        alpha = alpha_end + (alpha_start - alpha_end) * frac
        for _ in range(inner):
            new_pl = copy.deepcopy(cur_pl)
            if random.random()< 0.5:
                # swap move
                b1, b2 = random.sample(block_ids, 2)
                new_pl[b1][0], new_pl[b2][0] = new_pl[b2][0],
new_pl[b1][0]
                new_pl[b1][1], new_pl[b2][1] = new_pl[b2][1],
new_pl[b1][1]
            else:
                # slide move
                b = random.choice(block_ids)
                x,y,w,h = new_pl[b]
                if random.random()< 0.5:
                    dx = random.randint(-w//2, w//2)
                    nx = max(0, min(max_w - w, x + dx))
                    new_pl[b][0] = nx
                else:
                    dy = random.randint(-h//2, h//2)
                    ny = max(0, min(max_h - h, y + dy))
                    new_pl[b][1] = ny
            if not valid(new_pl):
                continue
            new_cost = cost(new_pl, alpha)
            delta = new_cost - cur_cost
            if delta < 0 or random.random()< math.exp(-delta / T):
                cur_pl = new_pl
                cur_cost = new_cost
                if cur_cost < best_cost:
                    best_pl = copy.deepcopy(cur_pl)
                    best_cost = cur_cost
        T *= decay

    # Format output
    placements = []
    for bid in block_ids:
        x,y,w,h = best_pl[bid]
        placements.append([bid, int(x), int(y), int(w), int(h)])
    return placements
```

**For Circuit n300:**

A two-phase floorplanner using skyline-based greedy placement followed by simulated annealing with an exponential HPWL-to-area weight schedule that starts balanced and transitions to area preference.

```python
import random, math

def floorplan(blocks, max_chip_size, nets):
    W, H = max_chip_size

    # new weight parameters
    alpha_start, beta_start = 0.5, 0.5
    alpha_end,  beta_end  = 0.8, 0.2

    # compute HPWL
    def compute_hpwl(place):
        centers = {bid: (x + w/2.0, y + h/2.0) for bid, (x, y, w, h) in
place.items()}
        hpwl = 0.0
        for net in nets:
            xs = [centers[b][0] for b in net if b in centers]
            ys = [centers[b][1] for b in net if b in centers]
            if xs and ys:
                hpwl += (max(xs) - min(xs)) + (max(ys) - min(ys))
        return hpwl

    # compute bounding area
    def compute_area(place):
        xs = [x for x, y, w, h in place.values()] + [x+w for x, y, w, h in
place.values()]
        ys = [y for x, y, w, h in place.values()] + [y+h for x, y, w, h in
place.values()]
        return (max(xs) - min(xs)) * (max(ys) - min(ys))

    # check overlap
    def no_overlap(bid, x, y, w, h, place):
        for obid, (ox, oy, ow, oh) in place.items():
            if obid == bid: continue
            if not (x + w <= ox or ox + ow <= x or y + h <= oy or oy + oh
<= y):
                return False
        return True

    # Phase 1: skyline-based greedy placement
    place = {}
    skyline = [(0, 0)] # frontier of possible placements
    # sort by max dimension descending
    sorted_blks = sorted(blocks, key=lambda b: -max(b[3], b[4]))
    for bid, _, _, w, h in sorted_blks:
        best, best_cost = None, float('inf')
        for (cx, cy) in skyline:
            if cx + w > W or cy + h > H: continue
            if not no_overlap(bid, cx, cy, w, h, place): continue
            c = alpha_start * compute_hpwl(place) + beta_start *
compute_area(place)
            def place(bid)
            if c < best_cost:
                best_cost, best = c, (cx, cy)
        if best is None:
            # fallback uniform scan
            step_x, step_y = max(1, W//20), max(1, H//20)
            for cx in range(0, W - w + 1, step_x):
                for cy in range(0, H - h + 1, step_y):
                    if no_overlap(bid, cx, cy, w, h, place):
                        best = (cx, cy)
                        break
                if best: break
        x0, y0 = best
        place[bid] = (x0, y0, w, h)
        # update skyline
        skyline.append((x0 + w, y0))
        skyline.append((x0, y0 + h))

    # Phase 2: simulated annealing with exponential weight schedule
    cur_hpwl = compute_hpwl(place)
    cur_area = compute_area(place)
    cur_cost = alpha_start * cur_hpwl + beta_start * cur_area
    T0 = max(cur_cost, 1.0)
    T, T_end, cooling = T0, 1e-3, 0.9
    iter_per_T = max(100, len(blocks) * 15)
    ids = [b[0] for b in blocks]

    while T > T_end:
        ratio2 = (T / T0) ** 2
        alpha = alpha_end + (alpha_start - alpha_end) * ratio2
        beta = beta_end + (beta_start - beta_end) * ratio2
        for _ in range(iter_per_T):
            if random.random()< 0.5:
                # swap two blocks
                b1, b2 = random.sample(ids, 2)
                p1, p2 = place[b1], place[b2]
                place[b1], place[b2] = p2, p1
                new_hpwl = compute_hpwl(place)
                new_area = compute_area(place)
                new_cost = alpha * new_hpwl + beta * new_area
                d = new_cost - cur_cost
                if d < 0 or random.random()< math.exp(-d / T):
                    cur_cost, cur_hpwl, cur_area = new_cost, new_hpwl,
new_area
                else:
                    place[b1], place[b2] = p1, p2
            else:
                # shift one block
                bid = random.choice(ids)
                x, y, w, h = place[bid]
                dx = random.randint(-w//3, w//3)
                dy = random.randint(-h//3, h//3)
                xn = min(max(0, x + dx), W - w)
                yn = min(max(0, y + dy), H - h)
                if no_overlap(bid, xn, yn, w, h, place):
                    continue
                old = place[bid]
                place[bid] = (xn, yn, w, h)
                new_hpwl = compute_hpwl(place)
                new_area = compute_area(place)
                new_cost = alpha * new_hpwl + beta * new_area
                d = new_cost - cur_cost
                if d < 0 or random.random()< math.exp(-d / T):
                    cur_cost, cur_hpwl, cur_area = new_cost, new_hpwl,
new_area
                else:
                    place[bid] = old
        T *= cooling

    # prepare placements
    placements = [[bid, x, y, w, h] for bid, (x, y, w, h) in place.items()]
    return placements
```

Figure 15: Specific details of the floorplanning heuristic algorithm designed for circuit n100, n200 and n300.

> **For Circuit ami33:**
>
> We use simulated annealing on block ordering with bottom-left decoding, minimizing a weighted cost = 2.0·HPWL + 0.5·area, starting at T0=10000, α=0.90, iter_per_T=3000.}

> **For Circuit ami49:**
>
> We initialize a grid placement then apply simulated annealing with random block swaps and position perturbations under non-overlap and boundary constraints to minimize HPWL and area.}

```
import random

def floorplan(blocks, max_chip_size, nets):
    # prepare block dims and degrees
    blk_dims = {b[0]: (b[3], b[4]) for b in blocks}
    bids = list(blk_dims.keys())
    deg = {b:0 for b in bids}
    for net in nets:
        for b in net:
            if b in deg:
                deg[b] += 1
    cw, ch = max_chip_size

    def compute_hpwl(placements):
        ctr = {b[0]:(b[1]+b[3]/2.0, b[2]+b[4]/2.0) for b in placements}
        tot = 0.0
        for net in nets:
            xs = [ctr[b][0] for b in net if b in ctr]
            ys = [ctr[b][1] for b in net if b in ctr]
            if xs and ys:
                tot += (max(xs)-min(xs)) + (max(ys)-min(ys))
        return tot

    def compute_area(placements):
        xs = [b[1] for b in placements] + [b[1]+b[3] for b in placements]
        ys = [b[2] for b in placements] + [b[2]+b[4] for b in placements]
        return (max(xs)-min(xs))*(max(ys)-min(ys))

    def bottom_left_place(order):
        placed = []
        xs = {0}
        ys = {0}
        for bid in order:
            w,h = blk_dims[bid]
            best = (cw, ch)
            for x in xs:
                if x + w > cw: continue
                for y in ys:
                    if y + h > ch: continue
                    ok = True
                    for _, px, py, pw, ph in placed:
                        if not (x+w <= px or px+pw <= x or y+h <= py or
py+ph <= y):
                            ok = False
                            break
                    if ok and (y < best[1] or (y == best[1] and x < best[0])):
                        best = (x, y)
            px,py = best
            placed.append([bid, px, py, w, h])
            xs.add(px + w)
            ys.add(py + h)
        return placed

    # hill-climbing with random restarts
    best_place = None
    best_cost = float('inf')
    restarts = 10
    iters = 500
    # initial ordering by descending degree
    base_order = sorted(bids, key=lambda b:deg[b], reverse=True)

    for _ in range(restarts):
        curr_order = base_order[:]
        random.shuffle(curr_order)
        curr_place = bottom_left_place(curr_order)
        curr_hpwl = compute_hpwl(curr_place)
        curr_area = compute_area(curr_place)
        curr_cost = curr_hpwl + curr_area

        for __ in range(iters):
            i,j = random.sample(range(len(bids)),2)
            nei_order = curr_order[:]
            nei_order[i], nei_order[j] = nei_order[j], nei_order[i]
            nei_place = bottom_left_place(nei_order)
            nei_hpwl = compute_hpwl(nei_place)
            nei_area = compute_area(nei_place)
            nei_cost = nei_hpwl + nei_area
            if nei_cost < curr_cost:
                curr_order = nei_order
                curr_place = nei_place
                curr_cost = nei_cost
                if curr_cost < best_cost:
                    best_cost = curr_cost
                    best_place = curr_place

    # if no improvement, take last
    if best_place is None:
        best_place = curr_place

    # integer output
    placements = [[b[0], int(b[1]), int(b[2]), int(b[3]), int(b[4])] for b in
best_place]
    return placements
```

```
import random
import math
import time

def floorplan(blocks, max_chip_size, nets):
    random.seed(0)
    chip_w, chip_h = max_chip_size
    placements = {}
    x = y = row_h = 0
    for bid,_,_,w,h in blocks:
        if x + w > chip_w:
            x = 0
            y += row_h
            row_h = 0
        if y + h > chip_h:
            y = 0
        placements[bid] = [x, y, w, h]
        row_h = max(row_h, h)
        x += w

    def compute_hpwl(pl):
        centers = {bid: (pos[0]+pos[2]/2.0, pos[1]+pos[3]/2.0) for bid,pos
in pl.items()}
        hpwl = 0
        for net in nets:
            xs = [centers[bid][0] for bid in net]
            ys = [centers[bid][1] for bid in net]
            hpwl += (max(xs)-min(xs)) + (max(ys)-min(ys))
        return hpwl

    def compute_area(pl):
        xs = [pos[0] for pos in pl.values()] + [pos[0]+pos[2] for pos in
pl.values()]
        ys = [pos[1] for pos in pl.values()] + [pos[1]+pos[3] for pos in
pl.values()]
        return (max(xs)-min(xs))*(max(ys)-min(ys))

    def is_valid(pl):
        items = list(pl.items())
        for bid, (x1,y1,w1,h1) in items:
            if x1 < 0 or y1 < 0 or x1+w1 > chip_w or y1+h1 > chip_h:
                return False
        n = len(items)
        for i in range(n):
            _, (x1,y1,w1,h1) = items[i]
            for j in range(i+1, n):
                _, (x2,y2,w2,h2) = items[j]
                if not (x1 >= x2+w2 or x2 >= x1+w1 or y1 >= y2+h2 or y2
>= y1+h1):
                    return False
        return True

    best_pl = dict(placements)
    best_cost = compute_hpwl(placements) +
compute_area(placements)
    T = 1e4
    start = time.time()
    for _ in range(10000):
        pl2 = {k: v[:] for k,v in placements.items()}
        if random.random() < 0.5:
            a,b = random.sample(list(pl2.keys()), 2)
            pl2[a][0],pl2[b][0] = pl2[b][0],pl2[a][0]
            pl2[a][1],pl2[b][1] = pl2[b][1],pl2[a][1]
        else:
            a = random.choice(list(pl2.keys()))
            w,h = pl2[a][2],pl2[a][3]
            pl2[a][0] = random.randint(0, chip_w-w)
            pl2[a][1] = random.randint(0, chip_h-h)
        if not is_valid(pl2):
            continue
        hp = compute_hpwl(pl2)
        ar = compute_area(pl2)
        cost = hp + ar
        d = cost - best_cost
        if d < 0 or random.random() < math.exp(-d/T):
            placements = pl2
            if cost < best_cost:
                best_cost = cost
                best_pl = dict(pl2)
        T *= 0.9995
        if time.time() - start > 1.9:
            break

    return [[bid, int(pos[0]), int(pos[1]), pos[2], pos[3]] for bid,pos in
best_pl.items()]
```

Figure 16: Specific details of the floorplanning heuristic algorithm designed for circuit ami33 and ami49.

### A.6.1 FOR THE N10 CIRCUIT.

The algorithm encodes three simple, human-interpretable rules: (1) prioritize highly connected blocks by initializing order with descending net-degree (mirrors designers placing hot-spots first), (2) use a bottom-left greedy packing heuristic to produce tight, low-waste floorplans (a classic manual/layout strategy), and (3) refine ordering via local swap hill-climbing optimizing a combined HPWL + area objective (a simple local improvement step designers would perform). These rules are intuitive and explainable.

### A.6.2   FOR THE N30 CIRCUIT.

Core idea : the planner decodes an ordering into a deterministic bottom-left packing, scores floorplans by HPWL plus a weighted area term, and explores orderings with simulated annealing using swap/insert moves and probabilistic acceptance. These components are directly interpretable and mirror human heuristics (place greedily to minimize gaps, trade off wirelength vs. area, and perform global-local randomized search). The fixed weight $\lambda$ compresses a designer's multi-objective tradeoff into a single scalar, so rules are transparent but simplify richer constraints (timing, routing, aspect ratios).

### A.6.3   FOR THE N50 CIRCUIT.

Core idea : place high-degree (highly connected) blocks first, greedily choose site candidates minimizing a combined HPWL–area cost, then refine with tabu-swapped positions to escape local minima. Each step maps to intuitive designer moves (prioritize nets, pack bottom/left-like sites, evaluate wirelength vs. area, and perform targeted swaps), so rules are transparent and human-aligned.

### A.6.4   FOR THE N100 CIRCUIT.

Core idea: initialize with a simple next-fit row packing (tall-first), score floorplans by a weighted sum of HPWL and bounding area, then explore via simulated annealing using swaps and small position perturbations with probabilistic acceptance. Each component maps directly to intuitive designer moves—pack greedily to reduce waste, trade off wirelength vs. area with explicit weights, and perform randomized local/global moves to escape local minima—so the discovered rules are transparent and align well with human layout heuristics.

### A.6.5   FOR THE N200 CIRCUIT.

Core idea: start with a simple tall-first row packing, evaluate floorplans by HPWL plus a dynamically weighted area term, then explore with simulated annealing using swaps (global) and slides (local). Each step corresponds to intuitive designer actions—place large/tall modules early, trade off wirelength vs. chip area via an explicit scalar, and perform localized/global adjustments to escape local minima—so the discovered rules are transparent and align well with common human heuristics.

### A.6.6   FOR THE N300 CIRCUIT.

Core idea: place large modules first using a skyline-style greedy search that evaluates candidate sites by a weighted HPWL–area criterion, then refine via simulated annealing with an evolving weight schedule $(\alpha, \beta)$ that shifts emphasis between wirelength and area; swap and small-shift moves mirror human local/global adjustments. The components and weight schedule are transparent and correspond closely to common designer heuristics.

### A.6.7   FOR THE AMI33 CIRCUIT.

Core idea: seed placement order by net-degree (place highly connected blocks earlier), decode an order via a deterministic bottom-left packing (tight, gap-minimizing), and improve order with local random swaps under a combined HPWL+area cost. These rules are simple, transparent and match common designer heuristics (prioritize high-degree modules, pack greedily, and perform local swaps to reduce wirelength/area).

### A.6.8   FOR THE AMI49 CIRCUIT.

Core idea: start with a simple row-based greedy packing, score floorplans by HPWL + bounding-box area, then perform randomized local search (swaps and random relocations) with Metropolis-style acceptance and temperature decay to escape local minima. These steps are directly interpretable and mirror common designer moves (greedy packing, trade off wirelength vs. area, and iterative stochastic refinement).

## A.7 CASE STUDY.

We now provide case studies designed to demonstrate the practical application and effectiveness of the AutoFloorplan optimization process, including the raw algorithms and code, violated constraint error, textual loss, textual gradients, textual updates, and textual gradient-guided repair algorithms and code.

We present a concise and traceable analysis demonstrating that code editing directly implements the repair guidance of the LLM's "textual gradient".

Mapping of each gradient item $\rightarrow$ where it appears in this case and why it is a faithful, interpretable implementation

- Gradient 1 — "Replace circle proxy repulsion with true rectangular separation."

Implementation: $New\_code$ (section "true rectangular repulsion + overlap penalty"). The code computes $rx = 0.5 * (wi + wj), ry = 0.5 * (hi + hj)$, then $ox = max(0, rx - abs(dx)), oy = max(0, ry - abs(dy))$ and applies forces only when $ox > 0$ && $oy > 0$.

Traceability: the overlap terms $ox, oy$ are explicit numeric diagnostics of $X/Y$ axis overlap (readable and testable), so one can directly verify the gradient's semantic goal (axis-aligned rectangle separation) by inspecting these variables each iteration.

- Gradient 2 — "Integrate hard legalization in the loop instead of post-clipping."

Implementation: $New\_code$ (section "Inline BL legalization (nudge)"). After each FD update the code builds bottom-left coordinates $bl[...]$, sorts $by(y, x)$, and performs incremental nudges against $settled$ blocks to resolve overlaps.

Traceability: the nudge loop and $settled$ list provide a readable sequence of per-block adjustments (you can log each push) so the reviewer can follow the exact steps that convert soft FD positions into a legal layout.

- Gradient 3 — "Add an overlap-penalty to the objective."

Implementation: $New\_code$ (in repulsion loop) introduces $K = K0 * tau$ and uses $(\beta * ox + K * ox)$ in the computed separation forces.

Traceability: $K$ is a scalar that increases with $tau$; its multiplication with $ox$ produces a measurable extra force proportional to overlapped area. This is a direct, auditable realization of the suggested soft penalty.

- Gradient 4 — "Dynamically schedule $dt, \beta$, and anneal spring strength."

Implementation: $New\_code$ (adaptive scheduling): $tau = t/(max\_iter - 1)$, $\beta = \beta0 * (1 - tau)$, $dt = dt0 * (1 - tau)$, and $K = K0 * tau$.

Traceability: the time schedule is parametric and explicit; logging $dt, \beta$, and $K$ per iteration yields a clear temporal trace showing early strong repulsion and later fine tuning as recommended by the gradient.

- Gradient 5 — "Improve initial packing (best-fit-decreasing / guillotine)."

Implementation: $New\_code$ (initialization): a best-fit-decreasing guillotine packer using $free\_rects$ and waste minimization.

Traceability: the chosen free rectangle and resulting placements are explicit (the $free\_rects$ structure and selected $best\_rect$), so the reviewer can follow how the initial state was changed to reduce downstream overlap pressure.

Each gradient item produces explicit variables/structures in code ($ox, oy, K, tau, \beta, dt, free\_rects$, $settled$) that are human-readable and can be logged or inspected at runtime. Therefore the repair guidance from the LLM is not merely high-level prose — it is translated into deterministic, testable code constructs whose evolution across iterations can be traced and audited.

## Example (Part 1)

**Raw Algorithm:**

We propose a force-directed continuous placement modeling blocks as repelling charges and nets as attracting springs to iteratively reduce HPWL and overall area under boundary constraints.

**Raw Code:**

```
import random, math
def floorplan(blocks, max_chip_size, nets):
    chip_w, chip_h = max_chip_size
    # 1) Initial next-fit bottom-left packing (by descending area)
    placements = {}
    x_cursor = y_cursor = line_height = 0
    for bid, _, _, w, h in sorted(blocks, key=lambda b: -(b[3]*b[4])):
        if x_cursor + w > chip_w:
            x_cursor = 0
            y_cursor += line_height
            line_height = 0
        placements[bid] = [x_cursor, y_cursor, w, h]
        x_cursor += w
        line_height = max(line_height, h)
    # 2) Initialize centers
    centers = {bid: [x + w/2.0, y + h/2.0]
               for bid, (x, y, w, h) in placements.items()}
    # 3) Force-directed iteration
    alpha = 0.2   # spring (attraction) strength
    beta = 5000.0  # repulsion strength
    gamma = 100.0  # boundary force strength
    dt = 0.01     # time step
    for _ in range(300):
        # zero out forces
        forces = {bid: [0.0, 0.0] for bid in placements}
        # attractive forces along nets
        for net in nets:
            pts = [centers[bid] for bid in net if bid in centers]
            if len(pts) < 2: continue
            cx = sum(p[0] for p in pts)/len(pts)
            cy = sum(p[1] for p in pts)/len(pts)
            for bid in net:
                if bid in centers:
                    dx = cx - centers[bid][0]
                    dy = cy - centers[bid][1]
                    forces[bid][0] += alpha * dx
                    forces[bid][1] += alpha * dy
        # repulsive forces between blocks
        bids = list(centers.keys())
        for i in range(len(bids)):
            bi = bids[i]
            xi, yi = centers[bi]
            ri = max(placements[bi][2], placements[bi][3])
            for j in range(i+1, len(bids)):
                bj = bids[j]
                xj, yj = centers[bj]
                rj = max(placements[bj][2], placements[bj][3])
                dx = xi - xj
                dy = yi - yj
                dist2 = dx*dx + dy*dy + 1e-6
                min_dist = (ri + rj)/2.0
                if dist2 < min_dist*min_dist:
                    f = beta / dist2
                    fx = f * dx
                    fy = f * dy
                    forces[bi][0] += fx
                    forces[bi][1] += fy
                    forces[bj][0] -= fx
                    forces[bj][1] -= fy
        # boundary forces
        for bid in centers:
            x, y = centers[bid]
            if x < 0:
                forces[bid][0] += gamma * (-x)
            if x > chip_w:
```

## Example (Part 2)

（*Continued from Part 1*）

```
            forces[bid][0] -= gamma * (x - chip_w)
        if y < 0:
            forces[bid][1] += gamma * (-y)
        if y > chip_h:
            forces[bid][1] -= gamma * (y - chip_h)
    # update centers and clip to valid region
    for bid in centers:
        centers[bid][0] += dt * forces[bid][0]
        centers[bid][1] += dt * forces[bid][1]
        w, h = placements[bid][2], placements[bid][3]
        cx = max(w/2.0, min(centers[bid][0], chip_w - w/2.0))
        cy = max(h/2.0, min(centers[bid][1], chip_h - h/2.0))
        centers[bid] = [cx, cy]
    # 4) Finalize integer bottom-left corners
    result = []
    for bid, _, _, _, _ in blocks:
        cx, cy = centers[bid]
        w, h = placements[bid][2], placements[bid][3]
        x = int(max(0, min(chip_w - w, cx - w/2.0)))
        y = int(max(0, min(chip_h - h, cy - h/2.0)))
        result.append([bid, x, y, w, h])
    return result
```

**Textual Loss：**

You should evaluate the floorplanning algorithm generated by the model.

> **User Query**.
> The desired goal is: HPWL is 46000, area is 1200000.

> Below is an algorithm for floor planning which has some errors, please analyze the reasons for the errors and give modifications to make the code work properly and optimize HPWL and Area to the maximum.

> **Error**:Existence block exists overlap!

> **Do NOT generate a response to the query. be concise.**

**Textual Gradient：**

Here are several targeted ways to sharpen the current routine so that you eliminate "existence block overlap" failures and drive HPWL/area toward the targets:

1) Replace the Circle‐Proxy Repulsion with True Rectangular Separation
   Problem: Using $r = max(w,h)/2$ treats blocks as circles, so slender rectangles can still interpenetrate without triggering a force.
   Fix: At each iteration compute the axis‐aligned overlap in X and Y:
   $$ox = max(0, (w_1/2 + w_2/2) - |dx|),$$
   $$oy = max(0, (h_1/2 + h_2/2) - |dy|).$$
   Only when both ox>0 and oy>0 is there an overlap. Then apply a separation force proportional to (ox, oy) along the appropriate axes (using sign(dx), sign(dy)).
   Benefit: Guarantees that any genuine rectangle overlap generates a restoring force, closing the gap between purely physical modeling and floorplan constraints.

## Example (Part 3)

（*Continued from Part 2*）

2) Integrate Hard Legalization in the Loop Instead of Post‑Clipping
   Problem: You simply clip centers to [0,chip_w]×[0,chip_h] at each step (and at the end), but that can slice modules into one another or push them outside.
   Fix: After each FD update, run a lightweight BL "nudge" pass: sort modules by increasing y (then x) and for each block, if it overlaps any settled block or violates boundary, push it just enough in +x or +y until the violation is resolved.
   Benefit: Keeps the layout overlap‑free throughout the FD process, preventing wild oscillations or cumulative illegal placements.

3) Add an Overlap‑Penalty to Your Objective (for Simulated Annealing or Hybrid)
   Problem: Pure FD doesn't directly minimize HPWL or area constraints; it only balances springs/repulsions.
   Fix: Define a soft‑penalty = $K \times$ (total overlapped area); add it to your "energy" and gradually increase K so the optimizer is forced to de‑overlap. Optionally embed small random swaps + acceptance criteria (SA‑style) every N iterations.
   Benefit: The floorplanner now has an explicit driving force to squeeze out overlaps while chasing lower HPWL.

4) Dynamically Schedule dt, $\beta$, and Anneal Spring Strength
   Problem: Fixed time‑step and weights mean you either under‑relax (never resolve overlaps) or overshoot (bouncing blocks).
   Fix: Start with larger $\beta$ (strong repulsion) and dt, then exponentially decay both over the 300 iters. Cool $\alpha$ (nets) more slowly so connectivity remains a gentle pull as the layout "freezes."
   Benefit: Early rapid de‑overlapping, later fine‑tuning of net‑length—all without manual tinkering.

5) Improve Initial Packing to Reduce Subsequent Overlap Pressure
   Problem: A naive next‑fit BL can leave large holes and force modules into deep overlap battles later.
   Fix: Use Best‑Fit‑Decreasing or a simple packing heuristic like largest‑area‑first with skyline or Guillotine cuts—so your FD starts from a more compact, less fragmentary layout.
   Benefit: Lowers initial overlaps and HPWL, speeding convergence and improving final area utilization.

By weaving true axis‑aligned repulsion, inline legalization, overlap penalties, adaptive scheduling, and a stronger init‑pack, you'll eradicate "existence block overlap" pathologies and move steadily toward the HPWL=46000, area=1.2M targets.

## Example (Part 4)

（*Continued from Part 3*）
**Textual Update:**

Given a text variable and its text gradient (modification suggestion), please optimize the algorithm and code in combination with the gradient.
**Gradient** Here are several targeted ways to sharpen the current routine so that you eliminate "existence block overlap" failures and drive HPWL/area toward the targets:

1) Replace the Circle‑Proxy Repulsion with True Rectangular Separation
    Problem: Using $r = \max(w,h)/2$ treats blocks as circles, so slender rectangles can still interpenetrate without triggering a force.
    Fix: At each iteration compute the axis‑aligned overlap in X and Y:
      $ox = \max(0, (w_1/2 + w_2/2) - |dx|)$,
      $oy = \max(0, (h_1/2 + h_2/2) - |dy|)$.
     Only when both $ox>0$ and $oy>0$ is there an overlap. Then apply a separation force proportional to $(ox, oy)$ along the appropriate axes (using $\text{sign}(dx)$, $\text{sign}(dy)$).
    Benefit: Guarantees that any genuine rectangle overlap generates a restoring force, closing the gap between purely physical modeling and floorplan constraints.
2) Integrate Hard Legalization in the Loop Instead of Post‑Clipping
    Problem: You simply clip centers to $[0,\text{chip\_w}]\times[0,\text{chip\_h}]$ at each step (and at the end), but that can slice modules into one another or push them outside.
    Fix: After each FD update, run a lightweight BL "nudge" pass: sort modules by increasing y (then x) and for each block, if it overlaps any settled block or violates boundary, push it just enough in +x or +y until the violation is resolved.
    Benefit: Keeps the layout overlap‑free throughout the FD process, preventing wild oscillations or cumulative illegal placements.
3) Add an Overlap‑Penalty to Your Objective (for Simulated Annealing or Hybrid)
    Problem: Pure FD doesn't directly minimize HPWL or area constraints; it only balances springs/repulsions.
    Fix: Define a soft‑penalty $= K \times$ (total overlapped area); add it to your "energy" and gradually increase K so the optimizer is forced to de‑overlap. Optionally embed small random swaps + acceptance criteria (SA‑style) every N iterations.
    Benefit: The floorplanner now has an explicit driving force to squeeze out overlaps while chasing lower HPWL.
4) Dynamically Schedule dt, $\beta$, and Anneal Spring Strength
    Problem: Fixed time‑step and weights mean you either under‑relax (never resolve overlaps) or overshoot (bouncing blocks).
    Fix: Start with larger $\beta$ (strong repulsion) and dt, then exponentially decay both over the 300 iters. Cool $\alpha$ (nets) more slowly so connectivity remains a gentle pull as the layout "freezes."
    Benefit: Early rapid de‑overlapping, later fine‑tuning of net‑length—all without manual tinkering.

## Example (Part 5)

（*Continued from Part 4*）

By weaving true axis‐aligned repulsion, inline legalization, overlap penalties, adaptive scheduling, and a stronger init‐pack, you'll eradicate "existence block overlap" pathologies and move steadily toward the HPWL=46000, area=1.2M targets.

**The Algorithm and Code** ······

**Other Information**First, describe your new algorithm and main steps in one sentence. The description must be inside a brace {}. After the brace containing the description, provide *only* the Python code implementation. The code block must be between <FLOORPLANCODESTART> and <FLOORPLANCODEEND> and contain any necessary `import` statements starting directly, followed by the function definition `def floorplan(...)`. The function implementation must contain only valid Python code. Do not include the algorithm description or any other explanatory text within the code block itself. The function Please design a floorplanning function to select the optimal position for each block and complete the floorplan for VLSI aiming to minimize HPWL and area. should accept 3 input(s): 'blocks', 'max_chip_size', 'nets'. The function should return 1 output(s): 'placements'.

Input:
- blocks: A list of blocks. Blocks are represented as a list [[block_id1, initial_x, initial_y, width, height], [block_id2, initial_x, initial_y, width, height], ...], [list(b) for b in blocks]. All values are integers. initial_x and initial_y might be placeholders (e.g., 0) if no initial placement is given.
- max_chip_size: A list representing the chip's maximum boundary dimensions [max_width, max_height]. Both values are integers.
- nets: A list of nets. Each net is a list containing the block_ids connected by that net.
Output:
- placements: A list of blocks, where each list represents the final placement of a block in the format [[block_id1, final_x, final_y, width, height], [block_id2, final_x, final_y, width, height], ...], [list(b) for b in blocks]. All values must be integers. The list should contain placements for all input blocks.

Constraints and Objectives:
1. No Overlap: Placed blocks must not overlap each other. The overlap area between any two distinct blocks must be zero.
2. Boundary Adherence: All placed blocks must be entirely within the chip maximum boundaries defined by chip_size (i.e., 0 <= x < chip_width and 0 <= y < chip_height for the bottom-left corner, and x + width <= chip_width and y + height <= chip_height for the top-right corner).
3. Minimize HPWL: The primary objective is to minimize the total Half‐Perimeter Wire Length (HPWL) calculated based on the final placement and the provided nets.
4. Minimize area: The main goal is to minimize the area occupied by the final layouts of blocks with the formula area=(max(blocks(x))-min(blocks(x)))*(max(blocks(y))-min(blocks(y))).

Do not give any additional explanations outside the initial description brace and the final code block.

## Example (Part 6)

（*Continued from Part 5*）

**Textual Gradient-Guided Repair Algorithm：**

A hybrid force-directed and legalization-based floorplanning algorithm with true rectangular repulsion, inline bottom-left legalization, adaptive scheduling, overlap penalties, and a best-fit-decreasing initial pack.

**Textual Gradient-Guided Repair Code：**

```
import math
def floorplan(blocks, max_chip_size, nets):
    chip_w, chip_h = max_chip_size
    # 1) Initial best-fit-decreasing guillotine packing
    free_rects = [(0, 0, chip_w, chip_h)]
    placements = {}
    for bid, _, _, w, h in sorted(blocks, key=lambda b: -(b[3]*b[4])):
        best_rect = None
        best_waste = float('inf')
        for rect in free_rects:
            fx, fy, fw, fh = rect
            if w <= fw and h <= fh:
                waste = fw*fh - w*h
                if waste < best_waste:
                    best_waste = waste
                    best_rect = rect
        if best_rect:
            fx, fy, fw, fh = best_rect
            free_rects.remove(best_rect)
            x, y = fx, fy
            right = (fx + w, fy, fw - w, h)
            top = (fx, fy + h, fw, fh - h)
            if right[2] > 0 and right[3] > 0: free_rects.append(right)
            if top[2] > 0 and top[3] > 0: free_rects.append(top)
        else:
            x, y = 0, 0
        placements[bid] = [x, y, w, h]
    # 2) Initialize centers
    centers = {bid: [x + w/2.0, y + h/2.0]
                for bid, (x, y, w, h) in placements.items()}
    # 3) Force-directed with adaptive scheduling and overlap penalty
    max_iter = 300
    alpha0, beta0, gamma, dt0, K0 = 0.2, 5000.0, 100.0, 0.02, 1000.0
    for t in range(max_iter):
        tau = t / float(max_iter - 1) if max_iter > 1 else 1.0
        alpha = alpha0
        beta = beta0 * (1 - tau)
        dt = dt0 * (1 - tau)
        K = K0 * tau
        # zero forces
        forces = {bid: [0.0, 0.0] for bid in placements}
        # attractive forces (nets)
        for net in nets:
            pts = [centers[bid] for bid in net if bid in centers]
            if len(pts) < 2: continue
            cx = sum(p[0] for p in pts) / len(pts)
            cy = sum(p[1] for p in pts) / len(pts)
            for bid in net:
                if bid in centers:
```

## Example (Part 7)

（*Continued from Part 6*）

```
            dx = cx - centers[bid][0]
            dy = cy - centers[bid][1]
            forces[bid][0] += alpha * dx
            forces[bid][1] += alpha * dy
        # true rectangular repulsion + overlap penalty
        bids = list(centers.keys())
        for i in range(len(bids)):
            bi = bids[i]
            xi, yi = centers[bi]
            wi, hi = placements[bi][2], placements[bi][3]
            for j in range(i+1, len(bids)):
                bj = bids[j]
                xj, yj = centers[bj]
                wj, hj = placements[bj][2], placements[bj][3]
                dx, dy = xi - xj, yi - yj
                rx = 0.5*(wi + wj)
                ry = 0.5*(hi + hj)
                ox = max(0, rx - abs(dx))
                oy = max(0, ry - abs(dy))
                if ox > 0 and oy > 0:
                    sx = 1 if dx >= 0 else -1
                    sy = 1 if dy >= 0 else -1
                    fx = (beta * ox + K * ox) * sx
                    fy = (beta * oy + K * oy) * sy
                    forces[bi][0] += fx
                    forces[bi][1] += fy
                    forces[bj][0] -= fx
                    forces[bj][1] -= fy
        # boundary forces
        for bid in centers:
            x, y = centers[bid]
            w, h = placements[bid][2], placements[bid][3]
            if x - w/2.0 < 0:
                forces[bid][0] += gamma * ((w/2.0) - x)
            if x + w/2.0 > chip_w:
                forces[bid][0] -= gamma * (x + w/2.0 - chip_w)
            if y - h/2.0 < 0:
                forces[bid][1] += gamma * ((h/2.0) - y)
            if y + h/2.0 > chip_h:
                forces[bid][1] -= gamma * (y + h/2.0 - chip_h)
        # update centers
        for bid in centers:
            centers[bid][0] += dt * forces[bid][0]
            centers[bid][1] += dt * forces[bid][1]
            w, h = placements[bid][2], placements[bid][3]
            cx = max(w/2.0, min(centers[bid][0], chip_w - w/2.0))
            cy = max(h/2.0, min(centers[bid][1], chip_h - h/2.0))
            centers[bid] = [cx, cy]
    # 4) Inline BL legalization (nudge)
    bl = {}
    for bid in centers:
        bl[bid] = [centers[bid][0] - placements[bid][2]/2.0,
                   centers[bid][1] - placements[bid][3]/2.0]
    order = sorted(bl.keys(), key=lambda b: (bl[b][1], bl[b][0]))
    settled = []
    for bid in order:
        w, h = placements[bid][2], placements[bid][3]
        x, y = bl[bid]
        changed = True
        while changed:
```

## Example (Part 8)

（*Continued from Part 7*）

```
        while changed:
            changed = False
            for obid, ox, oy, ow, oh in settled:
                oxp = min(x+w, ox+ow) - max(x, ox)
                oyp = min(y+h, oy+oh) - max(y, oy)
                if oxp > 0 and oyp > 0:
                    if x + w + oxp <= chip_w:
                        x += oxp
                    else:
                        y += oyp
                    changed = True
                    break
            x = max(0, min(x, chip_w - w))
            y = max(0, min(y, chip_h - h))
            bl[bid] = [x, y]
            centers[bid] = [x + w/2.0, y + h/2.0]
            settled.append((bid, x, y, w, h))
    # 5) Finalize integer bottom-left corners
    result = []
    for bid, _, _, _, _ in blocks:
        w, h = placements[bid][2], placements[bid][3]
        x = centers[bid][0] - w/2.0
        y = centers[bid][1] - h/2.0
        result.append([bid, int(round(x)), int(round(y)), w, h])
    return result
```

A.8 FLOORPLANS VISUALIZATION

(a) Floorplans for Chip n10

(b) Floorplans for Chip n30

(c) Floorplans for Chip n50

(d) Floorplans for Chip n100

(e) Floorplans for Chip n200

(f) Floorplans for Chip n300

Figure 17: Some Floorplans Visualization examples.

