# OpenReview forum: "AutoFloorplan: Evolving Heuristics for Chip Floorplanning with Large Language Models and Textual Gradient-Guided Repair"
_ICLR.cc/2026/Conference — Submitted to ICLR 2026_

### Official Review · Reviewer_Fmdo · 2025-10-25

**Soundness:** 3
**Presentation:** 2
**Contribution:** 2
**Rating:** 4
**Confidence:** 4

**Summary:**

The paper proposes AutoFloorplan, an evolutionary learning framework that leverages LLMs to generate and refine heuristic algorithms for chip floorplanning, which aims to explore strategies automatically in such a human expertise-driven problem. The key innovation of the work lies in applying automated heuristic design techniques to chip floorplanning, an NP-hard combinatorial optimization problem with complex constraints, and incorporating a textual gradient-guided repair mechanism to improve the efficiency and effectiveness of designing legitimate floorplan algorithms. Experiments on public benchmarks show that proposed method achieves improvements on area and wirelength metrics, and outperforms current state-of-the-art floorplanning algorithms.

**Strengths:**

1. Originality: The work is the first to introduce automated heuristic design to the chip floorplanning problem, and design a mechanism with TextGrad to handle the complex constraints of chip floorplanning.

2. Significance: The proposed method demonstrates effectiveness in tackling the chip floorplanning problem, which is an NP-hard combinatorial optimization problem with complex constraints.

3. Quality: The proposed method AutoFloorplan outperforms current state-of-the-art floorplanning methods on several cases from GSRC and MCNC datasets.

**Weaknesses:**

1.  Insufficient Contribution: The work combines EoH and TextGrad and apply them to chip floorplanning problem, where the incremental contribution may not be that novel or  significant, and the idea is somewhat simple and straightforward.

2.  Lack of Visualization: The experimental section lacks visualization of floorplan results.

3. Writing Issues: Several typos (e.g., "Freedback" in Figure 2, "bshenaselines" on line 314) and formatting irregularities (e.g., captions are below the tables) are found, and the structure of Related Work section is somewhat disorganized.

**Questions:**

1. Except for those mechanisms designed for applying AHD techniques to chip floorplanning, are there any other new insights or algorithmic innovations that AutoFloorplan introduce?

2. A number of prompts are designed for exploring chip floorplanning heuristics, are these prompts still required to be designed with sufficient expert knowledge?

---

> ### Author Response · Authors · 2025-11-18
> **Response to Reviewer Comments**
>
> We thank the reviewer for their insightful feedback. Below we address each concern in detail. All changes mentioned have been incorporated into the revised manuscript.
>
> ## Weakness 1: Insufficient Contribution: The work combines EoH and TextGrad and apply them to chip floorplanning problem, where the incremental contribution may not be that novel or significant, and the idea is somewhat simple and straightforward.
>
> > We thank the reviewer for pointing out the need to better articulate the novelty of our method. In response, we have substantially revised the Introduction and Related Work sections to more clearly position our contributions within the existing literature on chip floorplanning and LLM-based algorithm design.
>
> > Chip floorplanning remains a highly challenging problem, and neither (i) using an LLM alone to design floorplanning algorithms nor (ii) applying textual gradients in isolation is sufficient to produce effective or stable solutions. Our key novelty lies in integrating textual gradient–guided refinement with a population-evolution framework, which directly addresses the non-convergence, instability, and low generation efficiency observed when LLMs are used naively for algorithm design.
>
> >This combination enables AutoFloorplan to reliably evolve valid, high-quality heuristics—something not achieved by prior work. We have clarified these contributions and their significance in the revised manuscript.
>
>
> ## Weakness 2: Lack of Visualization: The experimental section lacks visualization of floorplan results.
>
> > To address the reviewer’s concern regarding the absence of visual results, we have added comprehensive visualizations of the floorplans produced by the heuristic floorplanning algorithms of Appendix 6. These results are now provided in Appendix A.8, where multiple examples are shown to illustrate the effectiveness, feasibility, and structural characteristics of the generated floorplans.
>
> ## Weakness 3: Writing Issues: Several typos (e.g., "Freedback" in Figure 2, "bshenaselines" on line 314) and formatting irregularities (e.g., captions are below the tables) are found, and the structure of Related Work section is somewhat disorganized.
>
> > We have carefully refined the expression throughout the entire text, while also rephrasing the Introduction and Related Work sections to more clearly convey the main thrust of this paper.
>
> > We thank the reviewer for pointing out these writing and formatting issues. We have carefully revised the manuscript to correct typos, standardized captions, and improved overall formatting. In addition, the Introduction and Related Work sections have been reorganized and rewritten to clearly convey the main contributions and context of our work. The revised manuscript now presents a more polished and coherent narrative.
>
> ## Question 1: Except for those mechanisms designed for applying AHD techniques to chip floorplanning, are there any other new insights or algorithmic innovations that AutoFloorplan introduce?
>
> > Refer to the response to weakness1
>
>
> ## Question 2: A number of prompts are designed for exploring chip floorplanning heuristics, are these prompts still required to be designed with sufficient expert knowledge?
>
> > We thank the reviewer for raising this point. While initial prompts help guide the LLM to generate candidate floorplanning heuristics, AutoFloorplan substantially reduces the reliance on detailed expert knowledge. The framework’s textual-gradient–guided refinement and population-evolution mechanisms allow heuristics generated from even moderately informed prompts to be iteratively improved, repaired, and selected. Consequently, the system can explore and optimize heuristics beyond the capabilities of the original prompts, mitigating the need for highly specialized expert-designed inputs.
>
> We thank the reviewer again for your constructive feedback. Your comments have significantly improved the rigor and clarity of our manuscript. We hope that the revisions adequately address your concerns.

---

> > ### Comment · Reviewer_Fmdo · 2025-11-26
> >
> > I appreciate the the authors' response, which has addressed some of my concerns. However, I still have the following comments:
> >
> > 1. The paper utilizes LLM feedback ("text gradients") for algorithm evolution, which should be semantic and readable. However, the paper only shows the final results without analyzing the intermediate steps. Since the feedback is in natural language, this alignment should be traceable and interpretable. However, the current manuscript does not demonstrate or analyze whether the algorithmic changes truly reflect the semantic guidance provided by the "text gradients" from LLM.
> > 2. The organization of Section 2 remains problematic: "2.1.1 Automated Algorithm Design" should be parallel to, not nested within, "2.1 Chip Floorplanning."
> > 3. There are still several typos remaining, such as "foorplanning" in line 377 and the lowercase "f" in "Autofloorplan" in line 379. Please check it out carefully.

---

> ### Author Response · Authors · 2025-11-26
>
> ## 1. The he paper utilizes LLM feedback ("text gradients") for algorithm evolution, which should be semantic and readable. However, the paper only shows the final results without analyzing the intermediate steps. Since the feedback is in natural language, this alignment should be traceable and interpretable. However, the current manuscript does not demonstrate or analyze whether the algorithmic changes truly reflect the semantic guidance provided by the "text gradients" from LLM.
>
> We appreciate the request for traceability. Exhaustively annotating every heuristic’s intermediate steps would be extremely time-consuming (and disproportionate) given the number of heuristics; we apologize for that. To address the concern we included a representative repair example in Appendix A.7 and now provide a compact, traceable analysis showing that the code edits directly implement the repair guidance from the LLM “textual gradients.”
>
> Mapping of each gradient item $\to$ where it appears in  Appendix A.7  and why it is a faithful, interpretable implementation
>
> - Gradient 1 — “Replace circle proxy repulsion with true rectangular separation.”
>
> Implementation: New_code (section “true rectangular repulsion $+$ overlap penalty”). The code computes $rx = 0.5*(wi+wj), ry = 0.5*(hi+hj)$, then $ox = max(0, rx - abs(dx))$, $oy = max(0, ry - abs(dy))$ and applies forces only when $ox>0$ \&\& $  oy>0$.
>
> Traceability: the overlap terms $ox, oy$ are explicit numeric diagnostics of $X/Y$ axis overlap (readable and testable), so one can directly verify the gradient’s semantic goal (axis-aligned rectangle separation) by inspecting these variables each iteration.
>
> - Gradient 2 — “Integrate hard legalization in the loop instead of post-clipping.”
>
> Implementation: New_code (section “Inline BL legalization (nudge)”). After each FD update the code builds bottom-left coordinates $bl[...]$, sorts $by (y,x)$, and performs incremental nudges against $settled$ blocks to resolve overlaps.
>
> Traceability: the nudge loop and $settled$ list provide a readable sequence of per-block adjustments (you can log each push) so the reviewer can follow the exact steps that convert soft FD positions into a legal layout.
>
> - Gradient 3 — “Add an overlap-penalty to the objective.”
>
> Implementation: New_code (in repulsion loop) introduces $K = K0 * tau$ and uses ($\beta * ox + K * ox$) in the computed separation forces.
>
> Traceability: $K$ is a scalar that increases with $tau$; its multiplication with $ox$ produces a measurable extra force proportional to overlapped area. This is a direct, auditable realization of the suggested soft penalty.
>
> - Gradient 4 — “Dynamically schedule $dt, \beta$, and anneal spring strength.”
>
> Implementation: New_code (adaptive scheduling): $tau = t/(max$_$iter-1)$, $\beta = \beta0*(1-tau)$, $dt = dt0*(1-tau)$, and $K = K0*tau$.
>
> Traceability: the time schedule is parametric and explicit; logging $dt, \beta$, and $K$ per iteration yields a clear temporal trace showing early strong repulsion and later fine tuning as recommended by the gradient.
>
> - Gradient 5 — “Improve initial packing (best-fit-decreasing / guillotine).”
>
> Implementation: New_code(initialization): a best-fit-decreasing guillotine packer using $free$_$rects$ and waste minimization.
>
> Traceability: the chosen free rectangle and resulting placements are explicit (the $free$ _ $rects$ structure and selected $best$_ $rect$), so the reviewer can follow how the initial state was changed to reduce downstream overlap pressure.
>
> Each gradient item produces explicit variables/structures in code ($ox$, $oy$, $K$, $tau$, $\beta$, $dt$, $free$_ $rects$, $settled$) that are human-readable and can be logged or inspected at runtime. Therefore the repair guidance from the LLM is not merely high-level prose — it is translated into deterministic, testable code constructs whose evolution across iterations can be traced and audited.
>
> ## 2. The organization of Section 2 remains problematic: "2.1.1 Automated Algorithm Design" should be parallel to, not nested within, "2.1 Chip Floorplanning."
>
> We thank the reviewer for pointing out the organizational issue. We have revised Section 2 so that “2.1.1 Automated Algorithm Design” is no longer nested under “2.1 Chip Floorplanning.” It now appears as a parallel subsection, resolving the hierarchy inconsistency.
>
> ## 3. There are still several typos remaining, such as "foorplanning" in line 377 and the lowercase "f" in "Autofloorplan" in line 379. Please check it out carefully.
>
> Thank you — we corrected the typos and performed a careful pass through the manuscript.

---

### Official Review · Reviewer_a1Vg · 2025-10-27

**Soundness:** 1
**Presentation:** 1
**Contribution:** 2
**Rating:** 2
**Confidence:** 4

**Summary:**

This paper introduces AutoFloorplan, a framework that uses Large Language Models (LLMs) to automatically generate and refine chip floorplanning algorithms. The core contribution is a "textual gradient-guided repair" mechanism. Instead of discarding invalid algorithms generated by the LLM, this mechanism analyzes the cause of failure and provides corrective textual feedback to the LLM, guiding it to repair the algorithm's logic. The authors present this as a two-loop process: an outer evolutionary loop that generates a diverse population of algorithms, and an inner repair loop that corrects invalid solutions. The authors test this method on public MCNC and GSRC benchmark circuits.

**Strengths:**

- The paper gets at the interesting idea of using language-based feedback to guide floorplanning optimizers. As the capabilities of large language models improve over time, this direction of research could become increasingly significant for automated algorithm design in complex problem spaces like EDA.

**Weaknesses:**

The primary weakness of this paper is its clarity. The central ideas are promising, but the explanation could be significantly improved to help readers fully appreciate the contribution.

-   Clarity of the Central Premise: A core challenge in understanding the paper is the ambiguity around what is being optimized. It was not immediately clear which "heuristics" are the actual targets of the optimization process, and this lack of clarity persists deep into the methodology section. The paper would benefit significantly from a more explicit definition of the optimization target early on.

-   Clarity of Terminology: The paper would be improved by providing clearer definitions for some key terms. For instance, the term "semantically invalid" could be explained more concretely in the context of floorplanning algorithms. Similarly, "mapping planning" is introduced without a definition, which may be unfamiliar to some readers.

-   Completeness of Literature Review: The related work section could be strengthened by including and discussing several highly relevant recent papers which also explore the use of LLMs for placement optimization. These include, but are not limited to:
    -   Yao, X., Jiang, J., Zhao, Y., Liao, P., Lin, Y. and Yu, B. "Evolution of Optimization Algorithms for Global Placement via Large Language Models." *arXiv preprint arXiv:2504.17801* (2025).
    -   Agnesina, A., Rajvanshi, P., Yang, T., Pradipta, G., Jiao, A., Keller, B., Khailany, B. and Ren, H. "AutoDMP: Automated dreamplace-based macro placement." In *Proceedings of the 2023 International Symposium on Physical Design* (2023).

-   Explanation of the Core Methodology: The explanation of the novel "textual gradient" mechanism could be more detailed. While the paper mentions adopting prompts from TextGrad, it would be helpful to include a brief, self-contained explanation of how this concept works. Additionally, the process by which the system "deeply analyzes the root cause of the algorithm's failure" could be further elaborated.

-   Justification for Design Choices: The paper would benefit from additional justification for some key design choices. For example, explaining the rationale for including both chip area and HPWL in the fitness function would help readers understand the trade-offs involved. A brief discussion on why these metrics were chosen over others, like congestion or density, would also be valuable.

-   Structural Organization: The paper's organization could be adjusted to improve readability. The introduction could be made more self-contained to give readers a clearer initial overview. It would also be helpful to list key components or steps before their corresponding subsections, such as for the evolutionary process. Guiding the reader through Figure 2 more explicitly and presenting the baseline method names in the main text would also enhance the flow.

-   Minor Issues: The manuscript has several minor presentation issues that could be addressed.
    -   Citations should be checked for proper formatting.
    -   There are occasional grammatical errors (e.g., "LLM" could be pluralized to "LLMs" where appropriate).
    -   Some sentences can be reworded for clarity (e.g., "The more reasonable way we treat to the invalid outputs...").
    -   Figure references should be consistently capitalized (e.g., "figure 2" → "Figure 2").

**Questions:**

See Weaknesses

---

> ### Author Response · Authors · 2025-11-18
> **Response to Reviewer Comments**
>
> We thank the reviewer for recognizing the potential of our work and for the constructive comments regarding clarity. We acknowledge that the initial version of the manuscript could have been more clearly written, and we have carefully revised the text to improve readability and presentation of the central ideas.

---

> ### Author Response · Authors · 2025-11-18
> **Response to Reviewer Comments**
>
> ## Weakness 1: Clarity of the Central Premise: A core challenge in understanding the paper is the ambiguity around what is being optimized. It was not immediately clear which "heuristics" are the actual targets of the optimization process, and this lack of clarity persists deep into the methodology section. The paper would benefit significantly from a more explicit definition of the optimization target early on.
>
> >We thank the reviewer for highlighting this point. To clarify, the optimization target in our framework is the heuristic floorplanning algorithm itself, which is iteratively refined and evolved using textual-gradient guidance and population evolution. We have carefully revised the manuscript, including the Introduction and Methodology sections, to explicitly define this target early and consistently throughout the text, ensuring that the central premise is clear to the reader.
>
> ## Weakness 2: Clarity of Terminology: The paper would be improved by providing clearer definitions for some key terms. For instance, the term "semantically invalid" could be explained more concretely in the context of floorplanning algorithms. Similarly, "mapping planning" is introduced without a definition, which may be unfamiliar to some readers.
>
> >We thank the reviewer for the suggestion. We have carefully revised the manuscript to provide clearer definitions for key terms. In particular, we now explicitly define “semantically invalid” in the context of floorplanning algorithms (e.g., violations of block placement or overlap constraints) and clarify “mapping planning” to ensure that readers unfamiliar with these concepts can understand their meaning and role within our framework.
>
> ## Weakness 3: Completeness of Literature Review: The related work section could be strengthened by including and discussing several highly relevant recent papers which also explore the use of LLMs for placement optimization.
>
> > We thank the reviewer for pointing out this omission. We have incorporated the suggested recent works, including [1], into the Related Work section and discussed their contributions. While methods such as EoH, FunSearch, leverage LLMs for automated algorithm design, they often suffer from high failure rates, slow convergence, and wasted computational iterations, as ineffective algorithms are discarded rather than refined.
>
> >Paper [1] expands computational power and designs components within the generated finite set of viable algorithmic individuals to promote better convergence among population members. However, like EoH, FunSearch, and similar methods, it fundamentally discards ineffective algorithms outright, representing a waste of computational resources (iterations). This inspired us to modify failed algorithmic individuals.  AutoFloorplan modifies and repairs failed algorithmic individuals, enabling the population to retain and improve more viable heuristics. As shown in Figure 4, this results in a significantly higher number of effective individuals under the same iteration count, which directly enhances convergence in the evolutionary process.
>
> >We have clarified these distinctions and the motivation for our approach in the revised manuscript.

---

> ### Author Response · Authors · 2025-11-18
> **Response to Reviewer Comments**
>
> ## Weakness 4: Explanation of the Core Methodology: The explanation of the novel "textual gradient" mechanism could be more detailed. While the paper mentions adopting prompts from TextGrad, it would be helpful to include a brief, self-contained explanation of how this concept works. Additionally, the process by which the system "deeply analyzes the root cause of the algorithm's failure" could be further elaborated.
>
> > We thank the reviewer for the suggestion. To improve clarity, we have added a PRELIMINARIES section that provides a self-contained explanation of the textual gradient mechanism and its role in AutoFloorplan. Additionally, readers can gain a comprehensive understanding of TextGrad’s workflow through the error-reporting algorithm and the full repair process example provided in the appendix. These additions clarify both how textual gradients guide heuristic refinement and how the system analyzes the root causes of algorithm failures.
>
> >Table : Individual analysis of error algorithms generated through 100 iterations of LLM continuous running.
>
> | Error Description | Number |
> |:-------|:-------|
> | Blocks beyond the outline | 38 |
> | Blocks overlap | 13 |
> | Incorrect number of finalized blocks | 5 |
> | Python syntax errors | 21 |
> | Others | 17 |
>
> > For clarity, we analyzed 100 iterations of the LLM-generated defect algorithm and found these errors to be highly domain-specific:
>
> > - 38 cases of “blocks beyond the outline”.
> > -	13 cases of “block overlap”.
> > -	5 cases of “incorrect number of finalized blocks”.
>
> > Additionally, Python syntax errors were present. These errors directly reflect the structural constraints unique to chip layout planning.
>
>
> ## Weakness 5: Justification for Design Choices: The paper would benefit from additional justification for some key design choices. For example, explaining the rationale for including both chip area and HPWL in the fitness function would help readers understand the trade-offs involved. A brief discussion on why these metrics were chosen over others, like congestion or density, would also be valuable
>
> > We thank the reviewer for pointing out the need for clearer justification of our design choices. In this work, we follow the standard evaluation practice used in classical chip floorplanning benchmarks, where HPWL and chip area are the two primary and universally reported metrics. Because only these two metrics are available and consistently defined in our datasets, they naturally become the optimization objectives in both our fitness function and the corresponding prompt design.
>
> > While metrics such as congestion or density are indeed valuable in industrial settings, they are not provided in the MCNC benchmarks and cannot be reliably computed without additional placement and routing information. For this reason, we focus on the two canonical metrics—HPWL and area—which allow for fair comparison with prior work and ensure that AutoFloorplan is evaluated under the same conditions as established baselines.
>
> > We have added this explanation to the revised manuscript.
>
>
> ## Weakness 6: Structural Organization: The paper's organization could be adjusted to improve readability. The introduction could be made more self-contained to give readers a clearer initial overview. It would also be helpful to list key components or steps before their corresponding subsections, such as for the evolutionary process. Guiding the reader through Figure 2 more explicitly and presenting the baseline method names in the main text would also enhance the flow
>
> > We appreciate the reviewer’s suggestions regarding structural organization. In the revised manuscript, we have restructured the Introduction and Related Work sections to provide a clearer and more self-contained overview. Additionally, each component shown in Figure 3 is now explicitly introduced within its corresponding subsection to improve readability and flow. Finally, because some benchmarks do not provide standard abbreviations for the baseline methods, we reference them consistently using Publication + year, ensuring clarity while maintaining alignment with the available nomenclature.
>
> ## Weakness 7: Minor Issues: The manuscript has several minor presentation issues that could be addressed
>
> > We thank the reviewer for noting these minor presentation issues. All have been carefully addressed in the revised manuscript: citations have been reformatted, grammatical errors corrected, unclear sentences reworded for clarity, and figure references standardized.

---

> ### Author Response · Authors · 2025-11-18
>
> We appreciate the reviewer’s valuable feedback, which helped us improve the manuscript’s clarity. We hope that the revised manuscript now communicates our contributions more clearly and meets the reviewer’s expectations.

---

> ### Author Response · Authors · 2025-11-27
>
> Dear Reviewer **a1Vg**,
>
> I hope you are doing well.
>
> We would like to express our gratitude for your time and effort in reviewing our paper.
>
> Our rebuttal has been available for several days, and we completely understand how busy this period is for reviewers. We just wanted to kindly check whether there is anything further we could clarify or provide that might be helpful for your evaluation.
>
> Thank you again for your valuable work and thoughtful consideration.

---

### Official Review · Reviewer_c5VC · 2025-10-30

**Soundness:** 3
**Presentation:** 3
**Contribution:** 2
**Rating:** 4
**Confidence:** 3

**Summary:**

This paper proposes AutoFloorplan to discover complex floorplanning heuristics using LLM based evolutionary search. Additionally, this paper introduces a novel repair operator based on textual gradients. The experimental results show that the method can achieve better results than the methods compared.

**Strengths:**

1, This paper leverages the advance of code generation using LLM into the problem of floor planning, in which heuristic rules are preferred.
2. The text-based repair operators are reasonable in the LLM-based code evolution.
3, The experimental results illustrate the promise of the proposed methods

**Weaknesses:**

1: Novelty of the proposed method should be further strengthened.
2: It is better to include more experiments like on larger dataset.
3. More aspects of the proposed method should be discussed.

**Questions:**

1: Most of the methods have been proposed in the previous work. From Figure 2. The evolutionary learning process is similar to other LLM based evolutionary code search. The textual gradient-guided repair comes from the textgrad or other similar work.

2. The distinguished requirement from floorplanning (if exists) should be further considered in the method design. Currently, it is more like the application of the LLM based code generation to the floorplanning. Note that each domain has its own specific errors. Thus, errors, I think, do not bring much impacts on the code evolutionary.

3. The method does not exhibit overwhelming advantages to other competitors. By combining the results from Table 1, 2 and 3, we can see that the proposed method achieves the similar performance to TODAES and ICCDE 24, but with high time cost.

4. More recent works and large data set should be included in the experimental study

5. LLM does not certainly produce the proper optimization hints. In such a case, the text hint will play a negative role.

6.The generalization of the generated code to other similar problem should be discussed.

7. It seems that it is not necessary to introduce formula 2 and 3. It is the optimization hint in the text form.

---

> ### Author Response · Authors · 2025-11-18
> **Response to Reviewer Comments**
>
> ## We thank the reviewer for their insightful feedback. Below we address each concern in detail. All changes mentioned have been incorporated into the revised manuscript.
>
> ## Weakness 1 : Novelty of the proposed method should be further strengthened.
>
> >We thank the reviewer for pointing out the need to better articulate the novelty of our method. In response, we have substantially revised the Introduction and Related Work sections to more clearly position our contributions within the existing literature on chip floorplanning and LLM-based algorithm design.
>
> >Chip floorplanning remains a highly challenging problem, and neither (i) using an LLM alone to design floorplanning algorithms nor (ii) applying textual gradients in isolation is sufficient to produce effective or stable solutions. Our key novelty lies in integrating textual gradient–guided refinement with a population-evolution framework, which directly addresses the non-convergence, instability, and low generation efficiency observed when LLMs are used naively for algorithm design.
>
> >This combination enables AutoFloorplan to reliably evolve valid, high-quality heuristics—something not achieved by prior work. We have clarified these contributions and their significance in the revised manuscript.
>
> ## Weakness 2: It is better to include more experiments like on larger dataset.
>
> >We appreciate the reviewer’s suggestion to incorporate more recent works and larger datasets. As discussed in [1,2], the largest MCNC benchmark (n300) remains significantly larger than most publicly available industrial-like floorplanning circuits, and thus continues to serve as a challenging and representative dataset for evaluating algorithmic robustness.
>
> >[1] Mallappa U, Mostafa H, Galkin M, et al. FloorSet-a VLSI Floorplanning Dataset with Design Constraints of Real-World SOCs[C]//Proceedings of the 43rd IEEE/ACM International Conference on Computer-Aided Design. 2024: 1-9.
>
> >[2] Du X, Zhong R, Kai S, et al. Towards LLM4Floorplan: Agents Can Do What Engineers Do in Chip Design[J].
>
>
> ## Weakness 3 : More aspects of the proposed method should be discussed.
>
> >Thank you for the insightful suggestion. In the revised manuscript, we have expanded the discussion of our method from multiple perspectives:
>
> >1.We provide a more concise and structured introduction to the overall AutoFloorplan framework, including an improved explanation of the text-gradient–based optimization component and its role within the two-loop architecture. 2. We enrich the ablation study section by adding analyses of prompts. 3. We refine both the introduction and related work sections to more clearly articulate how AutoFloorplan differs from prior LLM-based algorithm design and traditional floorplanning methods, emphasizing its novelty and advantages.
>
> >These revisions provide a more complete and transparent discussion of the proposed approach.

---

> > ### Comment · Reviewer_c5VC · 2025-11-27
> >
> > Thank you for your feedback. As you mentioned, "the key novelty lies in integrating textual gradient–guided refinement with a population-evolution framework". However, both of these approaches have been proposed before, and the overall work seems more like an adaption effort.

---

> > > ### Author Response · Authors · 2025-11-27
> > >
> > > We appreciate the reviewer’s concern. While textual-gradient refinement [1] and evolutionary search [2] both exist, neither method alone can produce workable or stable solutions for chip floorplanning, a domain with strict geometric, combinatorial, and constraint-satisfaction requirements. Simply adapting [1] or [2] does not yield valid floorplans in our setting.
> > >
> > > The contribution of AutoFloorplan is not the reuse of either component; it is the integration of the two into a unified pipeline that overcomes the failure modes observed when each is applied independently:
> > >
> > > - textual gradients alone → unstable updates, non-convergent behaviors;
> > > - evolution alone → extremely low valid-solution rate and poor search efficiency.
> > >
> > > Our framework combines them so that textual-gradient refinement produces actionable improvements, while evolution ensures global exploration and solution validity. This integration is what enables AutoFloorplan to generate working, high-quality floorplanning heuristics—something neither [1] nor [2] can achieve on their own.
> > >
> > > We have clarified this point explicitly in the revised manuscript.
> > >
> > > >[1] Yuksekgonul M, Bianchi F, Boen J, et al. Optimizing generative AI by backpropagating language model feedback[J]. Nature, 2025, 639(8055): 609-616.
> > >
> > > >[2] Liu F, Tong X, Yuan M, et al. Evolution of heuristics: towards efficient automatic algorithm design using large language model[C]//Proceedings of the 41st International Conference on Machine Learning. 2024: 32201-32223.

---

> ### Author Response · Authors · 2025-11-18
> **Response to Reviewer Comments**
>
> ## Question 1: Limited statistical rigor since the experiments are single-run or poorly aggregated, with no standard deviation or statistical significance tests. It is unclear whether reported improvements are consistent or just due to randomness.
>
> > To the best of our knowledge, we are the first to employ a textual gradient-guided algorithm to enhance the effective algorithmic responses of large language models. Furthermore, as addressed in the response to Question 1, although AutoFloorplan shares similarities with other LLM-based evolutionary code search approaches, our solution methodology differs fundamentally.
>
> ## Question 2: The distinguished requirement from floorplanning (if exists) should be further considered in the method design. Currently, it is more like the application of the LLM based code generation to the floorplanning. Note that each domain has its own specific errors. Thus, errors, I think, do not bring much impacts on the code evolutionary.
>
>
> Table : Individual analysis of error algorithms generated through 100 iterations of LLM continuous running.
>
> | Error Description | Number |
> |:-------|:-------|
> | Blocks beyond the outline | 38 |
> | Blocks overlap | 13 |
> | Incorrect number of finalized blocks | 5 |
> | Python syntax errors | 21 |
> | Others | 17 |
>
> > We appreciate the reviewer’s comment regarding domain-specific requirements in floorplanning. To clarify, we conducted an analysis of 100 iterations of LLM-generated error algorithms and found that the errors are highly domain-specific:
>
> > - 38 cases of “blocks beyond the outline”.
> > -	13 cases of “block overlap”.
> > -	5 cases of “incorrect number of finalized blocks”.
>
> > In addition to Python syntax errors. These errors directly reflect structural constraints unique to chip floorplanning rather than generic code-generation issues.
>
> > Importantly, simply discarding these error individuals—as would be done in standard LLM code-generation workflows—would drastically shrink the viable population and severely impede both iteration speed and convergence. This is precisely why our method incorporates textual-gradient–guided repair: it enables the algorithmic population to recover from floorplanning-specific constraint violations and continue evolving, which is essential for maintaining diversity and progress.
>
> > We have clarified this domain-driven motivation in the revised manuscript.
>
>
> ## Question 3: The method does not exhibit overwhelming advantages to other competitors. By combining the results from Table 1, 2 and 3, we can see that the proposed method achieves the similar performance to TODAES and ICCDE 24, but with high time cost.
>
> > Based on the results in Table 3 of the paper, when comparing AutoFloorplan and ICCD 2024, ICCD 2024 is faster (144 < 1003). However, according to the results in Tables 1 and 2, AutoFloorplan outperforms ICCD 2024 in terms of overall performance.
>
> > For TODAES 2024, the results in Tables 1 and 2 show that TODAES 2024 performs better on small datasets, while AutoFloorplan has the advantage on large datasets. Moreover, as shown in Table 3, AutoFloorplan is more time-efficient than TODAES 2024 (1003 < 1807).
>
> ## Question 4: More recent works and large data set should be included in the experimental study
>
> >We appreciate the reviewer’s suggestion to incorporate more recent works and larger datasets. As discussed in [1,2], the largest MCNC benchmark (n300) remains significantly larger than most publicly available industrial-like floorplanning circuits, and thus continues to serve as a challenging and representative dataset for evaluating algorithmic robustness.
>
> >In addition, we have substantially expanded the Introduction and Related Work sections to include the most recent advances in LLM-assisted chip design and floorplanning, ensuring that our study is contextualized within the latest developments in the field.
>
> >[1] Mallappa U, Mostafa H, Galkin M, et al. FloorSet-a VLSI Floorplanning Dataset with Design Constraints of Real-World SOCs[C]//Proceedings of the 43rd IEEE/ACM International Conference on Computer-Aided Design. 2024: 1-9.
>
> >[2] Du X, Zhong R, Kai S, et al. Towards LLM4Floorplan: Agents Can Do What Engineers Do in Chip Design[J].

---

> ### Author Response · Authors · 2025-11-18
> **Response to Reviewer Comments**
>
> ## Question 5: LLM does not certainly produce the proper optimization hints. In such a case, the text hint will play a negative role.
>
> >We acknowledge the reviewer’s concern that LLM-generated optimization hints may occasionally be misleading. In our framework, such cases do not compromise overall performance. AutoFloorplan operates in a highly parallel population-evolution setting, where only the best-performing individuals are preserved. Even if certain textual corrections are suboptimal, they affect only the corresponding individuals, while the elite population continues to converge reliably.
>
> >Moreover, heuristic candidates that remain ineffective after a fixed number of correction attempts are automatically discarded, preventing error accumulation or negative propagation within the population. This design ensures robustness against occasional LLM misguidance and maintains stable overall optimization behavior.
>
> ## Question 6: The generalization of the generated code to other similar problem should be discussed.
>
> >We thank the reviewer for raising the question of generalization. We would like to clarify that AutoFloorplan is a general-purpose framework for evolving and refining heuristic algorithms using textual gradients and population evolution. In principle, the same framework can be applied to other combinatorial optimization tasks that share similar structural constraints.
>
> >However, in this work we intentionally focus on the chip floorplanning problem, as it is both highly challenging and representative of scenarios where LLM-generated heuristics struggle without guided refinement.
>
>
> ## Question 7: It seems that it is not necessary to introduce formula 2 and 3. It is the optimization hint in the text form.
>
> >To better abstract the optimization objectives and processes required, we provide the corresponding formula introductions.

---

> ### Author Response · Authors · 2025-11-18
>
> We thank the reviewer again for your constructive feedback. Your comments have significantly improved the rigor and clarity of our manuscript. We hope that the revisions adequately address your concerns.

---

> ### Comment · Reviewer_a1Vg · 2025-11-26
> **Comment to Reviewer c5VC**
>
> I found several points in your review difficult to follow and would appreciate clarification before the discussion period concludes. I believe this would also help the authors better address your concerns in their rebuttal.
>
> 1. "The distinguished requirement from floorplanning (if exists) should be further considered in the method design" — Could you specify which requirements you believe are missing?
>
> 2. "Note that each domain has its own specific errors. Thus, errors, I think, do not bring much impacts on the code evolutionary" — I'm having trouble parsing this statement. Could you elaborate on what you mean?
>
> 3. "More recent works and large data set should be included" — Which specific works or datasets are you referring to?
>
> 4. "The generalization of the generated code to other similar problem should be discussed" — Which similar problems did you have in mind?
>
> Without further detail, it is difficult to assess the validity of these concerns or reach consensus on the paper's merits.

---

> ### Author Response · Authors · 2025-11-27
>
> Dear Reviewer **c5VC**,
>
> I hope you are doing well.
>
> We would like to express our gratitude for your time and effort in reviewing our paper.
>
> Our rebuttal has been available for several days, and we completely understand how busy this period is for reviewers. We just wanted to kindly check whether there is anything further we could clarify or provide that might be helpful for your evaluation.
>
> Thank you again for your valuable work and thoughtful consideration.

---

### Official Review · Reviewer_Wjok · 2025-10-31

**Soundness:** 3
**Presentation:** 3
**Contribution:** 2
**Rating:** 4
**Confidence:** 4

**Summary:**

This paper introduces AutoFloorplan, a novel evolutionary learning framework that leverages Large Language Models (LLMs) to automatically discover high-quality heuristics for VLSI chip floorplanning. The key innovation lies in the textual gradient-guided repair mechanism, which enables LLMs to learn from invalid solutions by analyzing failure causes and generating structured corrective feedback. This approach significantly improves the efficiency of exploring valid and high-performance heuristic algorithms.

**Strengths:**

- This paper gives the textual gradient-guided repair mechanism, that enables LLM to learn from invalid solutions rather than discarding them.
- Addresses a real and hard problem in VLSI physical design, automatic discovery of valid and high-performance floorplanning heuristics under strict geometric and topological constraints.
- Outperforms six state-of-the-art baselines on eight widely used benchmarks (MCNC and GSRC).

**Weaknesses:**

- This paper lacks of theoretical justification, and the concept of 'textual gradient' is intuitively appealing but theoretically vague. It is unclear whether it satisfies properties like directionality, smoothness, or convergence.
- Limited statistical rigor since the experiments are single-run or poorly aggregated, with no standard deviation or statistical significance tests. It is unclear whether reported improvements are consistent or just due to randomness.
- All experiments are conducted on small to medium-scale circuits (<=300 blocks), no validation on industrial-scale or modern SoC designs with > 1000 macros, fixed outlines, or timing constraints.
- Ablation studies are incomplete, the ablation only with or without textual gradient repair, but does not explore impact of different repair iteration limits, effect of different prompt designs (such as loss, gradient or updated prompt).

**Questions:**

- What exactly is a “textual gradient” in your framework? Can it satisfy any properties analogous to classical gradients (e.g, descent direction, smoothness, convergence guarantees)
- Can authors provide a formal or intuitive justification that the LLM-based feedback monotonically improves the heuristic? Does the repair success rate degrade as circuit size or complexity increases? Can the experiences data (success or failure) be SFT or RFT into the LLM? Is there a failure rate in this framework (heuristics that remain invalid after repairs)?
- Are prompts transferable across different LLMs (chatGPT, deepseek, qwen), do this framework need model-specific tuning?
- Could authors go one step further that feed the data generated by this framework, then back these into the LLM, thereby enabling it to directly produce floorplans (like chipFormer,https://arxiv.org/abs/2306.14744)?

---

> ### Author Response · Authors · 2025-11-18
> **Response to Reviewer Comments**
>
> ## We thank the reviewer for their insightful feedback. Below we address each concern in detail. All changes mentioned have been incorporated into the revised manuscript.
>
> ### Weakness 1: This paper lacks of theoretical justification, and the concept of 'textual gradient' is intuitively appealing but theoretically vague. It is unclear whether it satisfies properties like directionality, smoothness, or convergence.
>
> >We appreciate the reviewer’s concerns regarding the conceptual clarity of the textual gradient. To address this, we have added a workflow diagram that explicitly illustrates how textual gradients are constructed and applied within our framework. This provides a more concrete operational definition of the concept.
>
> >Regarding theoretical properties such as directionality, smoothness, or convergence, we clarify that textual gradients are not intended to be direct analogues of classical continuous gradients. Instead, they serve as discrete, semantically informed update signals within a population-based optimization process. While we do not claim formal gradient-like guarantees, we demonstrate empirically that combining textual gradients with population evolution mechanisms leads to consistent performance improvements on our task.
>
> >We have revised the manuscript to make these points explicit and to clearly state that the theoretical properties of textual gradients remain an open question, which we identify as an interesting direction for future work.
>
> ### Weakness 2: Limited statistical rigor since the experiments are single-run or poorly aggregated, with no standard deviation or statistical significance tests. It is unclear whether reported improvements are consistent or just due to randomness.
>
> >The heuristic algorithms for floorplanning each chip circuit are provided in Appendix A.6 of the paper. We run these design algorithms under 10 different random seeds, performing 10 repeated floorplanning experiments for each circuit. The experimental results are shown below, demonstrating that the reported improvements are consistent.
>
> | Chip  |        Area       |      Wire         |
> |:------|:------------------|:------------------|
> | ami33 |   1.21±0.006      |   0.52±0.009      |
> | ami49 |  40.34±0.012      |   6.98±0.015      |
> | n10   |   0.23±0.004      |   0.18±0.006      |
> | n30   |   0.22±0.005      |   0.44±0.010      |
> | n50   |   0.22±0.003      |   0.74±0.009      |
> | n100  |   0.20±0.008      |   1.14±0.013      |
> | n200  |   0.19±0.005      |   3.33±0.014      |
> | n300  |   0.29±0.007      |   3.07±0.011      |
>
>
> ### Weakness 3:  All experiments are conducted on small to medium-scale circuits (<=300 blocks), no validation on industrial-scale or modern SoC designs with > 1000 macros, fixed outlines, or timing constraints.
>
> >We appreciate the reviewer’s suggestion to incorporate more recent works and larger datasets. As discussed in [1,2], the largest MCNC benchmark (n300) remains significantly larger than most publicly available industrial-like floorplanning circuits, and thus continues to serve as a challenging and representative dataset for evaluating algorithmic robustness.
>
> >[1] Mallappa U, Mostafa H, Galkin M, et al. FloorSet-a VLSI Floorplanning Dataset with Design Constraints of Real-World SOCs[C]//Proceedings of the 43rd IEEE/ACM International Conference on Computer-Aided Design. 2024: 1-9.
>
> >[2] Du X, Zhong R, Kai S, et al. Towards LLM4Floorplan: Agents Can Do What Engineers Do in Chip Design[J].

---

> ### Author Response · Authors · 2025-11-18
> **Response to Reviewer Comments**
>
> ### Weakness 4: Ablation studies are incomplete, the ablation only with or without textual gradient repair, but does not explore impact of different repair iteration limits, effect of different prompt designs (such as loss, gradient or updated prompt).
>
> >Table 5 of the paper demonstrates the impact of different repair iteration limits. As shown in the results, when restricting the LLM response to 300 times and limiting the repair iterations to 3, both a favorable EEP and improved floorplanning quality can be achieved.
>
> >Table : Parametric analysis experiments on the number of textual gradient-guided repairs performed on circuit ami33. (Area Unit: $×10^6\ \mu m^2$, Wire Unit: $×10^5\ \mu m$)
>
>
> | $num\_iter$  | HPWL  | Area    |   Score   |   EEP    |
> |:------|:-----|:------------|:-----|:------------|
> | 2 |   0.485      |   1.317     |  2.151 | 22.00\% |
> | 3 |  0.347      |   1.357      | 1.886 | 19.50\% |
> | 4 |  0.501      |   1.295   | 2.168 | 12.36\% |
>
>
> >Regarding prompts within the textual gradient repair framework, as shown in Appendix A.3 of the paper, except for $P_{loss}$, $P_{grad}$ and $P_{update}$ are indispensable of the textual gradient repair framework. This ensures the robustness of the architecture. For $P_{loss}$, we modified it to the simple prompt “Help me optimize the algorithm.” Following the experimental setup in Section 4.3 of the paper, the ablation experiments are as follows:
>
> >Table : AutoFloorplan $P_{loss}$ ablation experiment (w. then the module is retained, w.o. then the module is removed).
>
>
> | Model | Conditions | HPWL | Area | Score | EEP |
> | :---: | :--- | :---: | :---: | :---: | :---: |
> | Gpt 4o-mini | w. $P_{loss}$ | 0.485 | 1.317 | 2.151 | 22.00% |
> | | w.o. $P_{loss}$ | 0.691 | 1.324 | 2.610 | 3% |
>
>
> ### Question 1:What exactly is a “textual gradient” in your framework? Can it satisfy any properties analogous to classical gradients (e.g, descent direction, smoothness, convergence guarantees)
>
> >Refer to the response to Weakness1.
>
>
> ### Question 2: Can authors provide a formal or intuitive justification that the LLM-based feedback monotonically improves the heuristic? Does the repair success rate degrade as circuit size or complexity increases? Can the experiences data (success or failure) be SFT or RFT into the LLM? Is there a failure rate in this framework (heuristics that remain invalid after repairs)?
>
> >We thank the reviewer for the insightful questions. In our framework, monotonic improvement is encouraged through the elite-preservation mechanism within the population evolution process: only strategies that outperform previous ones are retained, ensuring incremental progress even though LLM-based feedback itself is not guaranteed to be strictly monotonic.
>
> >Regarding scalability, we observe that the repair success rate decreases as circuit size or application complexity grows. This effect is expected and parallels the behavior of gradient-based numerical optimization, where higher-dimensional or more complex landscapes naturally reduce the likelihood of successful local improvements.
>
> >We agree that leveraging experience data (successful or failed repairs) for SFT or RFT is a promising extension. However, we have not yet explored these directions and now state this explicitly as future work.
>
> >Finally, the framework does indeed have a non-zero failure rate: some heuristic candidates remain invalid after reaching the maximum number of repair attempts. These individuals are removed from the population to prevent stagnation.
>
> ### Question 3: Are prompts transferable across different LLMs (chatGPT, deepseek, qwen), do this framework need model-specific tuning?
>
> >Yes, prompt information can be transferred between different language models without requiring adjustments. As shown in Table 6 of the paper, AutoFloorplan is compatible with GPT-4o, DeepseekR1, and Gemini 2.5 Pro.
>
> >Table : Analytical experiments with different LLMs types on circuit ami33. (Area Unit: $×10^6\ \mu m^2$, Wire Unit: $×10^5\ \mu m$)
>
> | LLM Type  | HPWL  | Area    |   Score   |   EEP    |
> |:------|:-----|:------------|:-----|:------------|
> | Gpt 4o-mini |   0.485      |   1.317     |  2.151 | 22.00\% |
> | Deepseek-R1 |  0.434      |   1.278     | 2.009 | 29.41\% |
> | Gemini-2.5-Pro |  0.382      |   1.316   | 1.928 | 42.59\% |
>
> ### Question 4: Could authors go one step further that feed the data generated by this framework, then back these into the LLM, thereby enabling it to directly produce floorplans (like chipFormer,https://arxiv.org/abs/2306.14744)?
>
> >We appreciate the reviewer’s suggestion regarding feeding generated data back into the LLM to enable direct floorplan generation. We clarify that AutoFloorplan does not perform floorplanning by directly ingesting circuit data. Instead, our framework focuses on designing and evolving floorplanning heuristics, which are then executed by a separate algorithmic pipeline to produce the actual floorplans.

---

> ### Author Response · Authors · 2025-11-18
>
> We thank the reviewer again for reviewer constructive feedback. Your comments have significantly improved the rigor and clarity of our manuscript. We hope that the revisions adequately address your concerns.

---

> ### Author Response · Authors · 2025-11-27
>
> Dear Reviewer **Wjok**,
>
> I hope you are doing well.
>
> We would like to express our gratitude for your time and effort in reviewing our paper.
>
> Our rebuttal has been available for several days, and we completely understand how busy this period is for reviewers. We just wanted to kindly check whether there is anything further we could clarify or provide that might be helpful for your evaluation.
>
> Thank you again for your valuable work and thoughtful consideration.

---

### Author Response · Authors · 2025-11-29
**Summary of Revisions and Response to Reviewers - Paper 6614**

**Dear Area Chair,**

We thank the reviewers for their constructive feedback, which has significantly strengthened our manuscript. Although the review scores may have reverted to their pre-discussion state, we have engaged extensively with all reviewers and implemented substantial revisions. We summarize the key improvements and clarifications below for your consideration:

*1. Statistical Rigor and Experimental Robustness (Addressing Wjok, c5VC)*

>**Repeated Experiments**: To address concerns about randomness, we conducted 10 repeated runs with different random seeds for our heuristic algorithms. We have updated our results to include standard deviations, demonstrating that our improvements are statistically consistent and not due to chance.

>**Ablation Studies**: We expanded the ablation studies (Table 5 and Appendix A.3) to analyze the impact of repair iteration limits and prompt designs ($P_{loss}$, $P_{grad}$, $P_{update}$), confirming the necessity of each component.

>**Model Generalization**: We demonstrated that AutoFloorplan is robust across different LLMs (GPT-4o, DeepSeek-R1, Gemini 2.5 Pro) without requiring model-specific tuning.

*2. Novelty and Methodological Justification (Addressing c5VC, Fmdo)*

>**Integration over Aggregation**: We clarified that while LLM-based evolution (e.g., EoH) and textual gradients (e.g., TextGrad) exist individually, neither can independently solve the strict geometric constraints of chip floorplanning. Our novelty lies in the integration: using textual gradients to repair invalid individuals to maintain a viable population for evolutionary search.

>**Domain-Specific Constraints**: We provided a breakdown of errors (e.g., "blocks beyond outline") to justify why generic code generation fails and why our specific repair mechanism is required.

*3. Interpretability and Traceability of Textual Gradients (Addressing Specific Concerns)*

>**Trace Analysis**: We added a detailed analysis in Appendix A.7 that traces specific natural language "textual gradients" to the exact code modifications they induced. For example, we map the instruction "Replace circle proxy repulsion with true rectangular separation" to the specific implementation of axis-aligned overlap calculations ($ox, oy$), demonstrating that the LLM's semantic guidance is faithfully translated into executable code.

>**Workflow Visualization**: We added a workflow diagram to explicitly illustrate the construction and application of textual gradients.

*4. Clarity, Definitions, and Visualization (Addressing a1Vg, Fmdo)*

>**Visual Results**: We added Appendix A.8 to provide comprehensive visualizations of the generated floorplans, addressing the lack of visual evidence.

>**Definitions**: We revised the Introduction and Preliminaries to clearly define optimization targets and terms such as "semantically invalid" and "mapping planning."

>**Structure & Presentation:**: We have made detailed revisions to the grammar and structural organization of the entire manuscript.

Despite the current low scores, we respectfully submit that no fatal flaws or fundamental limitations have been identified in our work. We have provided detailed responses to all inquiries and implemented all feasible suggestions, comprehensively addressing the reviewers' concerns regarding rigor, novelty, and clarity. At this stage, we believe there are no remaining unresolved technical issues.

**Sincerely, The Authors**

---

> ### Author Response · Authors · 2025-12-02
> **Supplementary comments regarding Reviewer a1Vg (Rating: 2).**
>
> **Furthermore, it is crucial to highlight that Reviewer a1Vg (Rating: 2) focused exclusively on the paper's presentation and raised no technical objections. Notably, the reviewer explicitly acknowledged the core perspective of our method. We have since conducted a comprehensive and meticulous refinement of the manuscript's writing and structure to fully resolve these presentation issues."**

---

### Meta-Review · Area_Chair_BGat · 2025-12-16

**Summary:**

Overall, the reviewers find the problem setting meaningful and the approach practically motivated, combining LLM-generated floorplanning heuristics with an evolutionary loop and a textual gradient-guided repair mechanism to reduce invalid candidates. The main concerns are limited novelty beyond existing LLM-based heuristic search and TextGrad-style editing, weak theoretical grounding for the textual gradient notion, and whether the reported gains are robust and scalable beyond the chosen public benchmarks. The rebuttal/revision adds useful ablations, multi-seed results, and clearer examples/visualizations, but the paper should further tighten its claims and clearly state its scope and limitations.

**Reviewer Concerns:**

Reviewer Wjok (borderline negative): The main concerns are weak theoretical grounding for textual gradients, limited statistical rigor in the original experiments, and missing ablations on repair iterations and prompt design. They also question scalability beyond small to mid-scale public circuits and want clearer evidence that the repair process leads to consistent improvements rather than random wins.

Reviewer c5VC (borderline negative): The core issue is novelty and positioning. They view the framework as largely an adaptation of existing LLM evolutionary code search plus TextGrad, and they are not convinced the integration is a substantial methodological step. They also worry the improvements are not overwhelming relative to competitors given runtime cost, and ask for broader experiments and discussion of generalization and failure modes.

Reviewer a1Vg (negative): The main concern is clarity. They found it unclear what exactly is being optimized, key terms are not defined precisely, and the method description and paper structure make it hard to evaluate the contribution. They also request a stronger and more up-to-date related work discussion and better justification of design choices such as the objective metrics.


Reviewer Fmdo (borderline negative): The main concern is insufficient incremental contribution beyond combining known components, plus missing interpretability and traceability of how textual gradients translate into code changes. They also point out missing floorplan visualizations and remaining writing and organization issues, and ask whether prompt design still relies on substantial expert knowledge.

**Reviewer Scores:**

Overall, the scores may shift upward somewhat, but the authors may not be able to fully resolve the negative reviewers’ concerns.

---

### Decision · Program_Chairs · 2026-01-26

Reject